# Deep learning-based enhancement of epigenomics data with AtacWorks

Avantika Lal [1,3], Zachary D. Chiang[2,3], Nikolai Yakovenko[1], Fabiana M. Duarte [2], Johnny Israeli[1✉] & Jason D. Buenrostro [2✉]

ATAC-seq is a widely-applied assay used to measure genome-wide chromatin accessibility; however, its ability to detect active regulatory regions can depend on the depth of sequencing coverage and the signal-to-noise ratio. Here we introduce AtacWorks, a deep learning toolkit to denoise sequencing coverage and identify regulatory peaks at base-pair resolution from low cell count, low-coverage, or low-quality ATAC-seq data. Models trained by AtacWorks can detect peaks from cell types not seen in the training data, and are generalizable across diverse sample preparations and experimental platforms. We demonstrate that AtacWorks enhances the sensitivity of single-cell experiments by producing results on par with those of conventional methods using ~10 times as many cells, and further show that this framework can be adapted to enable cross-modality inference of protein-DNA interactions. Finally, we establish that AtacWorks can enable new biological discoveries by identifying active regulatory regions associated with lineage priming in rare subpopulations of hematopoietic stem cells.

[1] NVIDIA Corporation, Santa Clara, CA, USA. [2] Department of Stem Cell and Regenerative Biology, Harvard University, Cambridge, MA, USA. [3]These authors contributed equally: Avantika Lal, Zachary D. Chiang. ✉email: jisraeli@nvidia.com; jason_buenrostro@harvard.edu

Within a single cell, the eukaryotic genome is hierarchically organized to form a gradient of chromatin accessibility ranging from compact, repressive heterochromatin to nucleosome-free regions associated with increased gene expression. Assay for Transposase-Accessible Chromatin using Sequencing (ATAC-seq) leverages the Tn5 transposase to directly measure chromatin accessibility as a proxy for the relative activity of DNA regulatory regions across the genome[1]. ATAC-seq has been applied to identify the effects of transcription factors on chromatin, construct cellular regulatory networks, and localize epigenetic changes underlying diverse development and disease-associated transitions[2–4]. Recently, the development of single-cell ATAC-seq methods have made it possible to measure accessible chromatin in individual cells, enabling epigenomic analysis of rare cell types within heterogeneous tissues[5].

The ability to measure biologically-meaningful changes in accessible chromatin using ATAC-seq depends on both the signal-to-noise ratio and the depth of sequencing coverage. Technical parameters such as the overall quality of cells or tissues, the nuclei extraction method[6], or over-digestion of chromatin can result in attenuated measurements of accessibility. Importantly, these issues are exacerbated in single-cell experiments, where primary tissues may vary in quality and key cell types may be exceedingly rare.

Deep learning represents a potential tool to address these limitations, as it has been successfully used for problems such as denoising speech[7] and image restoration[8,9]. An earlier study demonstrated that simple convolutional neural networks can be used to denoise and call peaks from ChIP-seq data, but was optimized for broad peak calling of histone modifications[10]. Another recent study applied deep learning to predict chromatin accessibility in a rare pancreatic islet cell type[11], highlighting the need for a robust and generalizable method for the analysis of sparse ATAC-seq data.

Here, we introduce AtacWorks (https://github.com/clara-parabricks/AtacWorks)[12], a deep learning-based toolkit that takes as input a low-coverage or low-quality ATAC-seq signal, and denoises it to produce a higher-resolution or higher-quality signal. AtacWorks trains a model to accurately predict both chromatin accessibility at base-pair resolution (a coverage track), and the genomic locations of accessible regulatory regions (peak calls). We apply AtacWorks to subsampled low-coverage bulk ATAC-seq and show that AtacWorks improves the resolution of the chromatin accessibility signal and the identification of regulatory elements. Further, AtacWorks is able to denoise signal from cell types not present in the training set, demonstrating that our deep learning models learn generalizable features of chromatin accessibility. We use the same framework to denoise aggregated single-cell ATAC-seq from a small number of cells, and also to improve the signal-to-noise ratio in an ATAC-seq dataset with low signal-to-noise. We further show that Atac-Works can be adapted for cross-modality prediction of transcription factor footprints and ChIP-seq peaks from low-input ATAC-seq. Finally, we apply AtacWorks to single-cell ATAC-seq of hematopoietic stem cells (HSCs) to identify regulatory elements associated with rare lineage-primed subpopulations.

## Results

### A deep learning framework for denoising low-coverage data.
AtacWorks trains a deep neural network to learn a mapping between noisy, low-coverage or low-quality ATAC-seq data and matching high-coverage or high-quality ATAC-seq data from the same cell type. Given a noisy ATAC-seq signal track as input, a trained model performs two tasks: denoising at base-pair resolution (predicting an improved signal track) and peak calling (predicting the genomic location of accessible regulatory elements). Once this mapping is learned, it is saved as a model that can be applied to denoise and call peaks from similar low-coverage or low-quality datasets at any given region in the genome.

The network makes predictions for each base in the genome based on coverage values from a surrounding region spanning several kilobases (6 kb for the models presented here), but does not consider the DNA sequence itself, allowing it to generalize across cell types. AtacWorks uses the ResNet (residual neural network) architecture, which has been applied extensively for natural image classification and localization[13]. Our architecture consists of multiple stacked residual blocks, each composed of three convolutional layers and a skip connection that bypasses intermediate layers (Fig. 1a). These skip connections allow propagation of the input through the layers of the network to avoid vanishing gradients[13], enabling deeper and more accurate models to be trained. The model is trained using a multi-part loss function combining Mean Squared Error (MSE), 1 - Pearson Correlation, and Binary Cross-Entropy (BCE) losses (see Methods).

We used AtacWorks to train deep learning models with bulk ATAC-seq data from FACS-isolated human blood-derived cell types[2]. To do this, we obtained ATAC-seq datasets from four cell types (B cells, natural killer (NK) cells, CD4$^+$ and CD8$^+$ T cells) and sampled each to a depth of 50 million reads (25 million read pairs) to produce standardized clean (high-coverage) data. Peaks for each clean dataset were identified using MACS2[14] (see Methods) which is the standard peak caller for ATAC-seq data, despite not being developed specifically for that purpose. We then subsampled each clean dataset to multiple lower sequencing depths ranging from 0.2 million to 20 million reads (Supplementary Fig. 1). For each depth, we trained a model to take as input the low-coverage ATAC-seq signal and reconstruct both the clean ATAC-seq signal and peak calls.

To assess the generalizability of our method, we tested the performance of these models on ATAC-seq data from erythroblasts[2], which were not included in the training set. We first subsampled reads from erythroblasts to the same depths as the training data. For each sequencing depth, we then applied the trained model to the corresponding subsampled dataset to obtain a predicted high-coverage signal track and peak calls (Fig. 1b). By examining the resulting denoised tracks, we confirmed that AtacWorks identifies cell-type-specific peaks that were not present in the training data, including a region adjacent to erythroblast marker gene *GYPA*[2] (Fig. 1c). This suggests that our models are learning generalizable features of chromatin accessibility rather than cell-type specific patterns.

To quantitatively evaluate the denoised high-coverage signal tracks produced by AtacWorks, we compared them to a clean (50 million read) erythroblast signal. At all sequencing depths, the Pearson correlation, Spearman correlation, and MSE between the denoised and clean signal tracks were substantially greater than that between the noisy and clean signal, both within and outside accessible chromatin peaks (Fig. 1d, Supplementary Table 1, Supplementary Fig. 2). We further found that our method outperforms smoothing using linear regression based on these metrics (Supplementary Table 2). Next, we evaluated the peaks identified by AtacWorks from each sequencing depth, and found that both the Area Under the Precision-Recall Curve (AUPRC) and Area Under the Receiver-Operator Characteristic (AUROC) of peaks were superior to MACS2 called peaks from the same subsampled data (Fig. 1e, Supplementary Table 1, Supplementary Fig. 2). For this analysis, AtacWorks produced output data of quality equivalent to (on average) 2.6× the number of reads in the input data based on Pearson correlation, and 4.2× based on AUPRC (Supplementary Table 1).

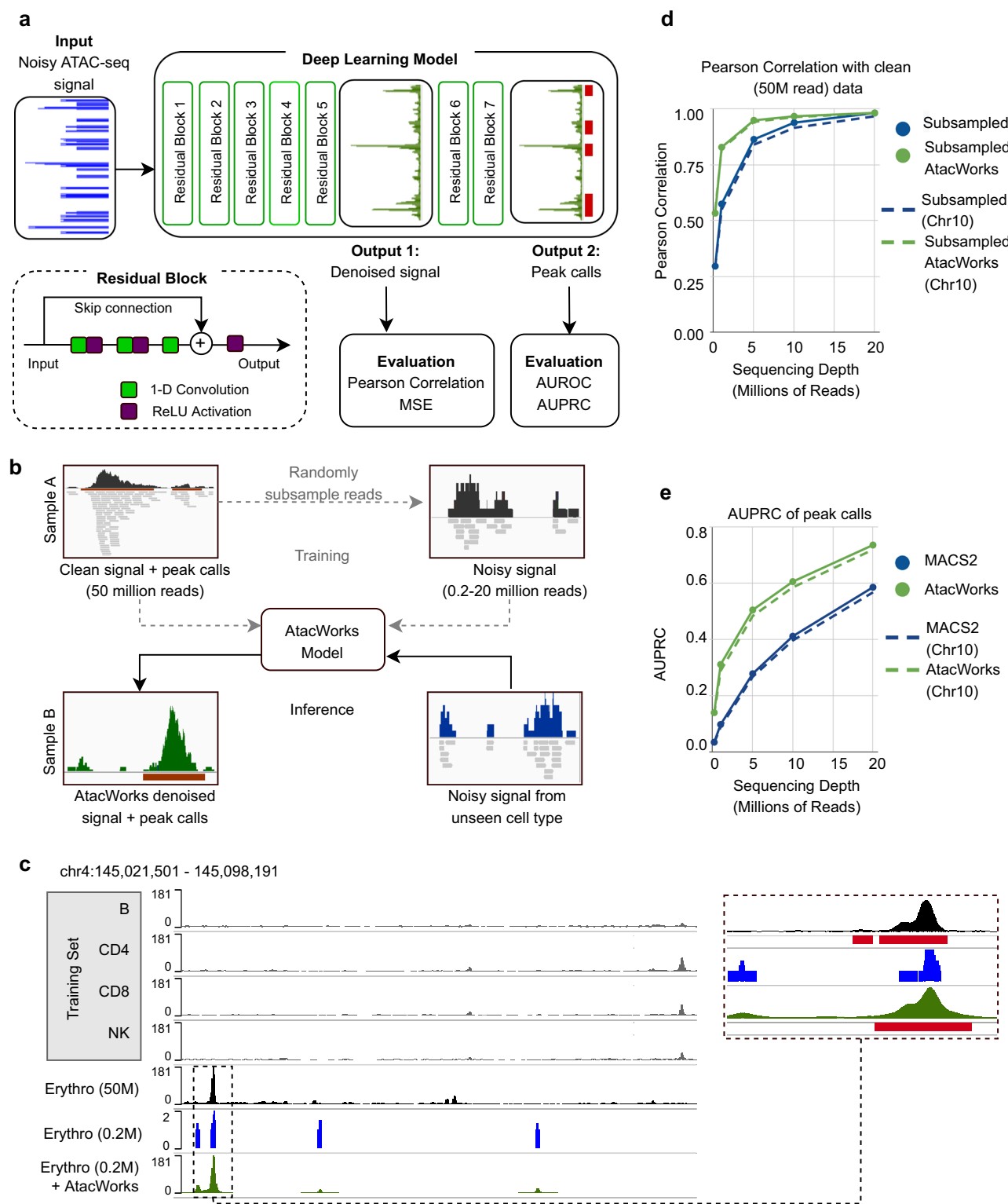

To show that the models are not simply learning features specific to the training set, we calculated performance metrics on chromosome 10, which was previously held-out from training, and obtained highly similar results to those computed on the whole genome (Figs. 1d and 1e, Supplementary Table 1). We also evaluated model performance specifically on differential peaks present in only either the training or test set, and found that

AtacWorks improves both the signal track accuracy and peak calling in these regions (Supplementary Table 1). Further, we found that the results were highly robust to different subsets of the training data used (Supplementary Table 3).

Since ATAC-seq is commonly applied to tissues containing a mixture of cell types, we sought to test whether our models could be applied to data of this nature. We found that a model trained

**Fig. 1 A deep learning approach to denoise ATAC-seq data. a** Schematic of the ResNet architecture. The zoomed-in region displays a residual block composed of 1-dimensional convolutional layers (green squares), nonlinear ReLU activation functions (purple squares), and a skip connection. **b** Schematic showing how to train and validate AtacWorks on subsampled bulk ATAC-seq data. Clean high-coverage bulk ATAC-seq data is subsampled to create noisy data. Matched pairs of clean and noisy data are used to train AtacWorks models, which are then applied to denoise and call peaks from subsampled noisy data derived from a different cell type. **c** ATAC-seq signal tracks near the erythroblast marker gene *GYPA*, for four cell types used to train an AtacWorks model (gray), high-coverage erythroblast data (50 million reads; black), and erythroblast data subsampled to 0.2 million reads before (blue) and after (green) denoising with AtacWorks. Red bars below the zoomed-in tracks show peak calls by MACS2 (for the 50 M and 0.2 M read tracks) and AtacWorks (for the denoised track). **d** Pearson correlation between a clean ATAC-seq signal track (50 million reads) and subsampled data for erythroblasts, before (blue) and after (green) denoising with AtacWorks. Solid lines show correlation over the genome; dotted lines show correlation over chromosome 10. **e** AUPRC for MACS2 (blue) and AtacWorks (green) showing their peak calling performance on subsampled data, using peaks called by MACS2 subcommands on the clean (50 million reads) signal track as ground truth. Solid lines show AUPRC over the genome; dotted lines show AUPRC over chromosome 10. AUPRC: Area Under the Precision-Recall Curve. AUROC: Area Under the Receiver Operating Characteristic. MSE: Mean Squared Error. ReLU: Rectified Linear Unit. Source data are provided as a Source Data file.

on FACS-isolated cell types from human blood was able to denoise subsampled low-coverage ATAC-seq data from a mixture of human cell types derived from the intestinal Peyer's Patch by the ENCODE consortium[15,16] (Supplementary Fig. 3, Supplementary Table 4). This suggests that our models are robust to data from mixtures of cell types, as well as varied experimental preparation of cells and tissues. However, we note that a model trained on three different ENCODE datasets produces better results on this task (Supplementary Table 4), suggesting that results may be improved when the training and test data are obtained from the same experimental protocol.

Another present challenge in adapting ATAC-seq to novel biological contexts is developing experimental protocols that optimize enrichment of open chromatin. To help address this issue, we applied AtacWorks to improve signal quality in ATAC-seq datasets with low signal-to-noise ratio. We trained a model to learn a mapping between paired high and low-quality ATAC-seq datasets from FACS-isolated human monocytes[2] (Supplementary Table 5, Methods). Both datasets had similar sequencing depth (~20 million reads); however, one had a higher signal-to-noise ratio estimated using the global enrichment of signal surrounding transcription start sites (TSSs). We then applied this trained model to denoise low-quality bulk ATAC-seq data of similar depth from erythroid cells. AtacWorks improved the enrichment at TSSs (Supplementary Fig. 4a), producing a signal track and peak calls more similar to those obtained from higher-quality data (Supplementary Fig. 4b, Supplementary Table 6).

Finally, we compared our method to a recent study[11] that also reported the use of a deep learning model that could perform either ATAC-seq denoising or peak calling. We implemented the U-Net model architecture reported in this study, and found that the ResNet architecture used in AtacWorks outperforms this model in denoising, peak calling, and runtime (Supplementary Note 1, Supplementary Table 7).

**AtacWorks enhances single-cell data from low numbers of cells**. To demonstrate our method is also adaptable to broad use cases of ATAC-seq, we applied AtacWorks to denoise data from a high-throughput single-cell ATAC-seq experiment. We first obtained droplet single-cell ATAC-seq (dscATAC-seq) data from bead-isolated human blood cells and aggregated single-cell chromatin accessibility profiles by cell type[17]. We selected two cell types (B cells and monocytes) from the dataset, and produced clean ATAC-seq signal tracks and peak calls by aggregating data over 2400 cells (~50 million reads) of each type. We then generated noisy ATAC-seq signals by randomly subsampling subsets of cells of each type, and trained AtacWorks models on the paired clean and noisy datasets (Fig. 2a). We randomly sampled 1 cell (~20,000 reads), 5 cells (~0.1 million reads), 10 cells (~0.2 million reads) or 50 cells (~1 million reads) for the low-coverage training

datasets. The resulting trained models improved signal track accuracy and peak calling from aggregated NK cells sequenced using the same protocol (Fig. 2b, Supplementary Table 8, Supplementary Table 9). Notably, AtacWorks improved the AUPRC of peak calls from 50 NK cells from 0.2048 to 0.7008, a result that MACS2 requires over 400 cells to obtain (Fig. 2b, Supplementary Table 8). Though we observed improved signal quality and peak calls for any number of cells, the results on 1 and 5 cell samples may be too noisy for downstream biological analysis, possibly due to single-cell heterogeneity not captured by the aggregate data used for training.

We then tested whether these models trained on dscATAC data from human blood could generalize to less-similar cell types. To do this, we obtained single-cell data from a mouse brain using the same dscATAC protocol[17]. We then applied the models trained on human blood to data aggregated from mouse pyramidal and oscillatory neurons. For both types of neurons, we observed that AtacWorks improved the signal track and peak calls, both overall and within cell-type specific peaks (Fig. 2c, d, Supplementary Data 1). This result demonstrates that AtacWorks is broadly applicable across both cell types and species.

Finally, because the previous experiment was limited to dscATAC data, we sought to investigate the generalizability of AtacWorks models to data from different single-cell platforms. To this end, we applied one of the previously-described AtacWorks models trained on dscATAC-seq data to human CD4+ T cells sequenced using a combinatorial indexing approach (dsciATAC-seq[17]), and observed improvements in both the signal track and peak calls (Supplementary Table 10). We also applied a similar model trained on dscATAC-seq data from human blood to macrophages from mouse primary lung tumors sequenced using the sciATAC-seq protocol[18]. Once again, we observed that the model trained on human dscATAC-seq data improved both signal track accuracy and peak calls (Supplementary Table 11). However, we note that a model trained on sciATAC-seq data from B cells and monocytes returned slightly better results on most metrics when applied to the same sciATAC-seq dataset from macrophages (Supplementary Table 11). Collectively, these results support AtacWorks as a highly generalizable tool to study single-cell ATAC-seq data.

**AtacWorks enables cross-modality predictions**. Seeing that AtacWorks accurately predicts denoised coverage at base-pair resolution, we sought to extend it for transcription factor footprinting[1,19]. Footprinting leverages the fact that transcription factors vary in how they bind to DNA, which allows binding events to be identified via a characteristic Tn5 insertion signature. Traditionally, footprinting requires over 100 million reads[19], prohibiting its widespread use. To test the feasibility of performing footprinting from low-input ATAC-seq, we obtained high-

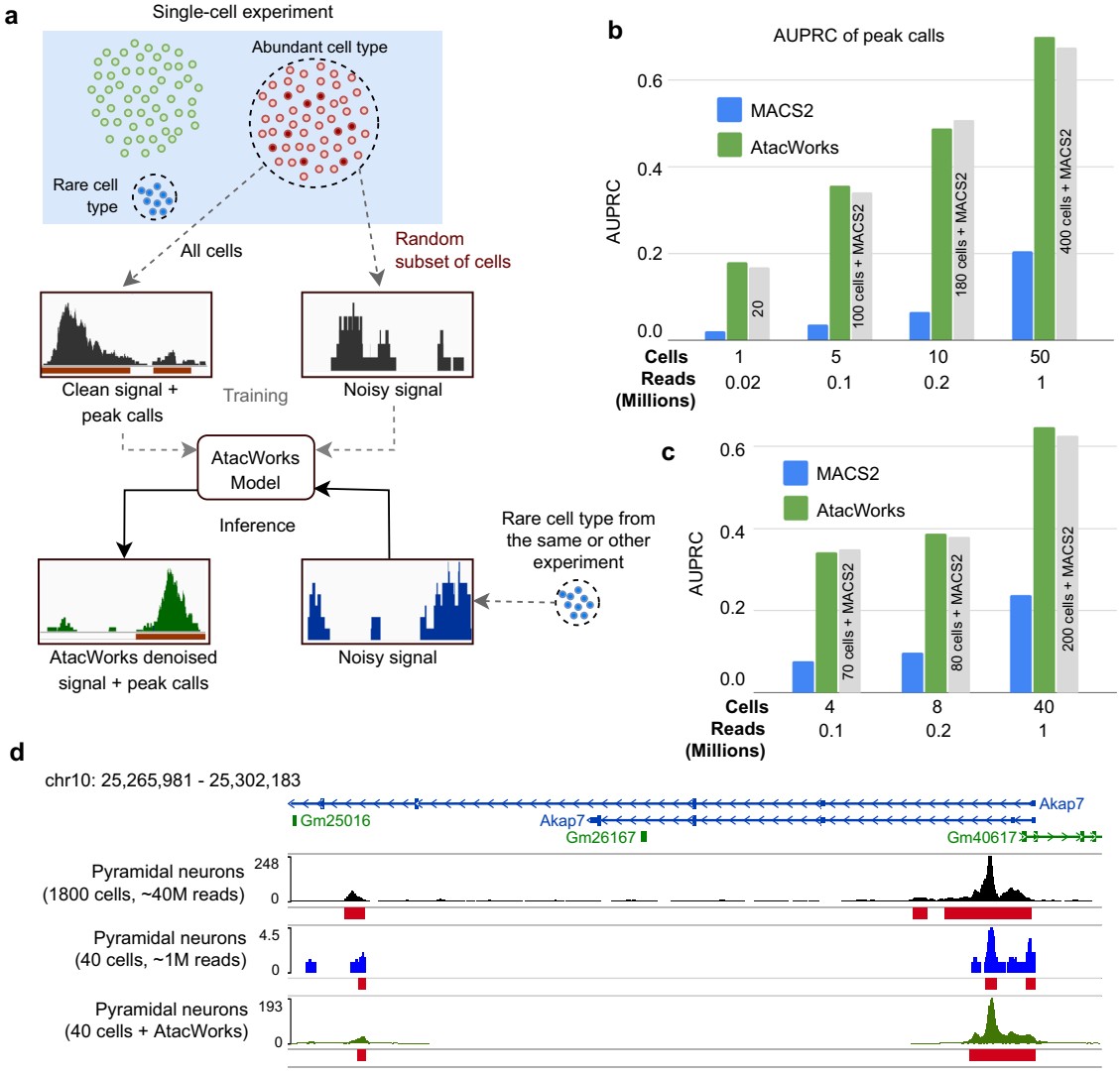

**Fig. 2 AtacWorks enhances single-cell data from low numbers of cells. a** Schematic showing the strategy to train and test AtacWorks on single-cell ATAC-seq data. A clean high-coverage ATAC-seq signal is obtained by aggregating data from all cells belonging to an abundant cell type. Data are aggregated over a randomly selected subset of these cells to obtain a noisy signal. Paired clean and noisy datasets are used to train an AtacWorks model. The model can be applied to denoise and call peaks from noisy aggregate data from small numbers of cells, either from the same experiment or a different experiment. **b** AUPRC of peak calls on aggregate single-cell ATAC-seq data from human natural killer (NK) cells. Peak calls were produced by MACS2 (blue) and AtacWorks (green) on noisy data aggregated over 1–50 cells. Gray bars show AUPRC of MACS2 on larger numbers of cells, to illustrate how many cells MACS2 requires to reach the same performance as AtacWorks. **c** AUPRC of peak calls on aggregate single-cell ATAC-seq data from mouse pyramidal neurons. Peak calls were produced by MACS2 (blue) and AtacWorks (green) on noisy data aggregated over 4, 8 or 40 cells. Gray bars show AUPRC of MACS2 on larger numbers of cells, to illustrate how many cells MACS2 requires to reach the same performance as AtacWorks. **d** ATAC-seq signal tracks near the *Akap7* gene, for single-cell mouse pyramidal neuron data (~40 million reads from 1800 cells; black), and neuron data subsampled to 40 cells (~1 million reads), before (blue) and after (green) denoising with AtacWorks. Red bars show peak calls by MACS2 (for the 1800 cell and 40 cell tracks) and AtacWorks (for the denoised track). AUPRC: Area Under the Precision-Recall Curve. Source data are provided as a Source Data file.

coverage (100 million reads) ATAC-seq data from FACS-sorted human blood cell types (multipotent progenitor cells, CD8[+] T cells, NK cells)[2] and reduced track smoothing to preserve transcription factor-specific patterns of Tn5 insertions (see Methods). We then downsampled these tracks to lower sequencing depths and trained a model for each depth, which we tested on data from similarly-processed HSCs. We evaluated the performance of these models on a set of 200 bp genomic regions spanning binding motifs for genome architectural protein CTCF. At all sequencing depths, AtacWorks improved the signal track spanning CTCF motifs in HSCs (Supplementary Table 12),

enhancing the characteristic footprint of CTCF binding (Supplementary Fig. 5).

Encouraged by these results, we reasoned we may adapt our method to directly predict ChIP-seq peaks from low-input ATAC-seq. Like footprinting, standard ChIP-seq protocols also require large quantities of input material (at least 10[7] cells), though this number has been reduced in certain contexts by recent technological developments[20]. To demonstrate the feasibility of cross-modality prediction, we trained AtacWorks models to learn a mapping from low-coverage aggregate dscATAC-seq signal to CTCF and H3K27ac (an active histone mark) ChIP-seq

signal and peak calls in the same cell type. For the prediction of CTCF ChIP-seq, we also supplied the model with the positions of CTCF binding motifs on both strands of the genome (see Methods). We trained models on noisy aggregate dscATAC-seq data from small numbers of B cells, and tested them on similarly-processed monocytes. For small numbers of cells ranging from 10 to 500, AtacWorks predicted CTCF and H3K27ac peak calls with surprisingly high concordance to ChIP-seq data from the same cell type (AUROC > 0.9 from 500 cells; Supplementary Fig. 6, Supplementary Data 2).

These cross-modality predictions demonstrate the potential for AtacWorks to generate multiple layers of information in single cells from one of the most commonly-used epigenomic assays, at no additional cost. It is generally experimentally challenging to make multiple measurements from the same cells, so this approach may be especially useful in cases where running multiple ChIP-seq experiments is infeasible due to time, reagents, sample availability, or biological variability. Though the models presented here tend to perform better on active histone marks (e.g., H3K27ac) or abundant architectural proteins (e.g., CTCF), these specific predictions may be useful for distinguishing active vs. poised enhancers[21] or characterizing changes in 3D genome structure across differentiation[22]. We anticipate future work will extend these capabilities to enable cross-modality inference of additional latent epigenetic states from a single experiment.

**AtacWorks enhances the resolution of single-cell studies.** Empowered by the improved resolution afforded by AtacWorks, we sought to investigate epigenetic changes underlying differentiation in rare cell subpopulations that cannot be experimentally isolated, and thus cannot be analyzed using traditional approaches. Previous single-cell studies of FACS-isolated bone marrow mononuclear cells (BMMCs) have observed epigenetic heterogeneity within immunophenotypically-defined cellular populations, suggesting that hematopoietic stem and progenitor cells lie along a continuum of differentiation states (Fig. 3a)[23,24]. In particular, HSCs are thought to include rare subpopulations of cells that are primed toward either the lymphoid or erythroid lineage[23,25,26]. Though single-cell ATAC-seq enables measurements of chromatin accessibility over aggregate genomic features, such as sets of transcription factor motifs[27] or the regions surrounding TSSs[27,28], with such granular lineage-primed states, there is typically not enough sequencing coverage to identify which specific regulatory regions are associated with each differentiation trajectory.

We reasoned we could use AtacWorks to identify sets of regulatory regions that are unique to lymphoid or erythroid-primed HSCs. First, we performed dscATAC-seq[17] on FACS-isolated HSCs to generate 9974 single-cell chromatin accessibility profiles (see Methods). To define lymphoid and erythroid differentiation trajectories, we collected published dscATAC-seq data from bead-enriched CD34+ cells and used a bulk reference-guided approach (see Methods) to project all single-cell profiles into a shared latent space, visualized using UMAP for dimensionality reduction (Fig. 3b, c). This analysis localized FACS-isolated HSCs to a region at the top of the projection. We then confirmed that HSCs localized in this region exhibited directional signal bias in transcription factor motif accessibility scores for the GATA2 motif (Fig. 3d) and smoothed gene accessibility scores for *MEF2C* (Fig. 3e), genes which have been implicated as markers of erythroid and lymphoid lineage priming respectively[24,29] (see Methods).

To generate high-resolution chromatin accessibility tracks of lineage-primed cells using our model, we selected three distant

samples of 50 similar HSCs each, representing putative populations of long-term, lymphoid and erythroid-primed HSCs (Fig. 3b). For each sample of 50 aggregated cells, we performed signal denoising using AtacWorks and visualized the denoised chromatin accessibility profiles near genes suggested to be indicators of lineage priming[24,29] (Fig. 3f). We observed considerable differences between the denoised tracks that could not be readily distinguished from the original low-coverage signal (Supplementary Fig. 7), including potential regulatory elements seemingly present in the lymphoid, but not the erythroid-primed cells (near *MEF2C*, *POU2F2*) and vice-versa (near *GATA1*, *GATA2*).

To assess the significance of these chromatin accessibility differences, we took 1000 random samples of 50 similar HSCs each and calculated a normalized mean and standard deviation of the coverage from the 1000 denoised tracks, allowing us to estimate *z*-scores for each regulatory region we observed in our denoised long-term HSC and lineage-primed samples (see Methods). We identified a total of 8590 significant regulatory regions surrounding genes associated with differential expression in CD34+ cells (Supplementary Data 3). To validate that these identified regulatory elements are associated with lineage-priming, we confirmed that the lymphoid-primed elements were more accessible in the CD34+ cells from lymphoid lineage (Supplementary Fig. 8a), while the erythroid-primed elements were more accessible in CD34+ cells from the erythroid lineage (Supplementary Fig. 8b). We also observed that the most differentially-accessible sequence motifs in these subsets of peaks included transcription factors crucial to differentiation, including E2F[30] and MYB families[31] (Supplementary Table 13). Altogether, these results demonstrate the unique capacity of deep learning to enhance the resolution of sparse single-cell ATAC-seq studies.

## Discussion

ATAC-seq has become a widely adopted tool for high-resolution characterization of the epigenome, providing insights into the mechanisms underlying gene expression changes associated with development, evolution, and disease. However, technical limitations in tissue quality, assay performance, and sequencing coverage constrain our ability to measure the full spectrum of chromatin states across the genome. These limitations also pertain to emerging single-cell ATAC-seq technologies, as cell types of interest are often difficult to experimentally isolate, and are present at low frequencies in heterogeneous contexts.

Here we present AtacWorks, an easy-to-use and generalizable toolkit to train and apply deep learning models to ATAC-seq data. Unlike previous deep learning methods for epigenomics, AtacWorks denoises ATAC-seq signal at base-pair resolution and simultaneously predicts the genomic location of accessible regulatory elements. The models we present here outperform existing approaches at both of these tasks, and moreover, are robust across cell and tissue types, individuals, and experimental protocols. AtacWorks is not provided with the DNA sequence as an input, which means it is agnostic to cell- or condition- specific correlations between chromatin accessibility and sequence motifs. Instead, the model learns features based on the shape of the coverage track, which generalize across datasets. In addition to generalization across different cell types, we also observed that our trained models can generalize to data from different species, experimental platforms, and quality levels. However, we observed that we could obtain slightly better results (e.g., AUPRC increase from 0.7332 to 0.7483) on a test dataset by using a model trained on more closely matched data (Supplementary Table 11), suggesting that there remain small benefits to matching training and test data when possible.

We also demonstrate that AtacWorks can be adapted for cross-modality prediction of transcription factor footprints and ChIP-

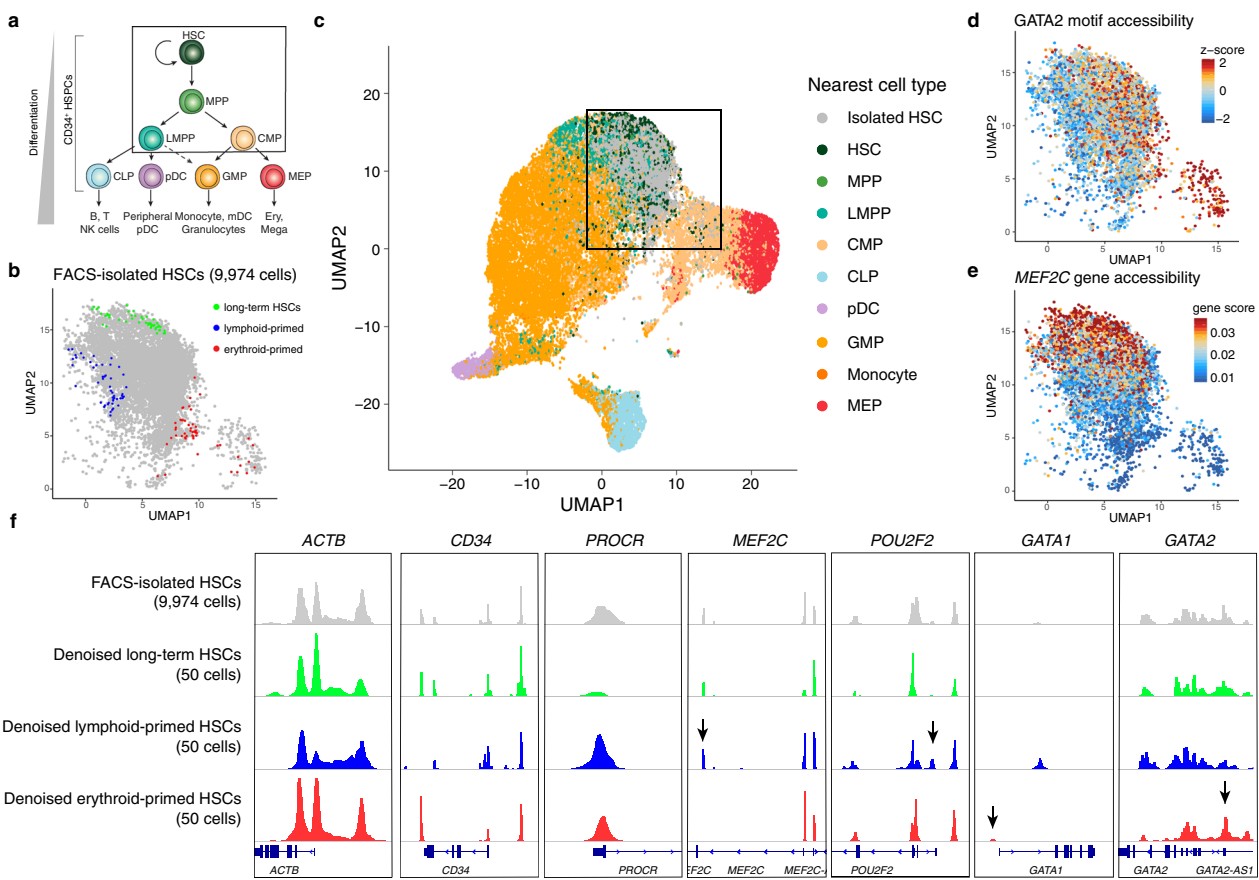

**Fig. 3 AtacWorks identifies differentially-accessible regulatory regions associated with lineage-primed hematopoietic stem cells. a** A schematic of the classical hierarchy of human hematopoietic differentiation. **b** A UMAP dimensionality reduction of single-cell ATAC-seq profiles from 9974 FACS-isolated hematopoietic stem cells (HSCs). The colored points represent three 50-cell subsamples, each generated by selecting a single cell and identifying its nearest neighbors in principal component space. **c** A combined UMAP dimensionality reduction of single-cell ATAC-seq profiles from HSCs shown in (**b**) and 28,505 published bead-enriched CD34+ bone marrow progenitor cells[17]. The bead-enriched CD34+ cells are colored by the most correlated cell type from a FACS-isolated single-cell ATAC-seq reference[24]. The box indicates the region containing FACS-isolated HSCs shown in (**b**), (**d**) and (**e**). **d** FACS-isolated HSCs colored by chromVAR transcription factor motif accessibility z-scores for GATA2. These scores represent enrichment or depletion of chromatin accessibility within peaks that contain the GATA2 motif (see Methods). **e** FACS-isolated HSCs colored by smoothed gene accessibility scores for *MEF2C*. These scores are a weighted sum of read counts within 10 kb of the *MEF2C* transcription start site, averaged over each cell's 50 nearest neighbors (see Methods). **f** Aggregate chromatin accessibility signal tracks surrounding genes implicated as markers of lineage priming[24,29] for all 9974 FACS-isolated HSCs and the three denoised 50-cell subsamples of HSCs shown in (**b**). The arrows denote select regulatory regions with significant differences in chromatin accessibility relative to a random background. HSC: hematopoietic stem cell. MPP: Multipotent progenitor. LMPP: lymphoid-primed multipotent progenitor. CMP: common myeloid progenitor. CLP: common lymphoid progenitor. pDC: plasmacytoid dendritic cell. GMP: granulocyte-macrophage progenitor. MEP: megakaryocyte-erythroid progenitor. Source data are provided as a Source Data file.

seq peaks from low-input ATAC-seq. As such, we anticipate this framework may be broadly useful for other deep learning applications in genomics, such as DNase, MNase, ChIP-seq, and the recently-developed method CUT&RUN[20], which has comparable high-throughput single-cell adaptations[32,33].

Finally, the robustness and speed of AtacWorks enable its application to high-throughput single-cell ATAC-seq datasets of heterogeneous tissues. We show that our method can be used on small subsets of rare lineage-priming cells to denoise signal and identify accessible regulatory regions at previously-unattainable genomic resolution. Based on these advancements, we anticipate that AtacWorks will broadly enhance the utility of epigenomic assays, providing a powerful platform to investigate the regulatory circuits that underlie cellular heterogeneity.

## Methods
**Data preprocessing**. BAM files for bulk ATAC-seq were downsampled to a fixed number of reads using SAMtools v.1.9[34]. For paired-end sequencing data, both

reads in a read pair were retained (e.g., 100,000 read pairs were selected to obtain a total of 200,000 sequencing reads). For CTCF footprinting experiments, downsampling was repeated independently five times to produce five times the amount of training data. This was done to ensure that the model received enough training data, as only a small fraction of the genome was used for training in these experiments.

For single-cell ATAC-seq experiments, a number of cells of the chosen cell type were randomly selected and all reads from those cells were extracted from the BAM file via cell barcodes. This random sampling of cells was repeated independently five times due to the sparsity of the input single-cell ATAC-seq data.

To identify the exact location of Tn5 insertions with base-pair resolution, each ATAC-seq read was converted to a single genomic position corresponding to the first base pair of the read. Previous work has demonstrated that the Tn5 transposase inserts adapters separated by 9 bp, so reads aligning to the + strand were offset by +4 bp, while reads aligning to the - strand were offset by −5 bp[1]. Each cut site location was extended by 100 bp in either direction, except for transcription factor footprinting experiments where each cut site was extended by 5 bp in either direction. The bedtools genomecov function (v2.26.0)[35] was then used to convert the list of locations into a genome coverage track containing the ATAC-seq signal at each genomic position.

To call peaks from clean and noisy signal tracks, MACS2 (v2.2.7) subcommands bdgcmp and bdgpeakcall were run with the ppois parameter and a -log10(p value)

cutoff of 3. BED files with equal coverage over all chromosomes were provided as a control input track.

**Input data for AtacWorks**. Deep learning models were trained using one or more pairs of matching ATAC-seq datasets. Each pair consisted of two ATAC-seq datasets from the same sample or cell type: a clean dataset of high sequencing coverage or quality, and a noisy dataset of lower coverage or quality. Unless indicated otherwise, low-coverage datasets were generated computationally by randomly subsampling a fraction of reads or cells from the high-coverage dataset.

Models were given three inputs for each such pair of datasets:

1. a signal track representing the number of sequencing reads mapped to each position on the genome in the noisy dataset.
2. a signal track representing the number of sequencing reads mapped to each position on the genome in the clean dataset.
3. the genomic positions of peaks called by MACS2 on the clean dataset.

Models learned a mapping from (1) to both (2) and (3); in other words, from the noisy signal track, they learned to predict both the clean signal track, and the positions of peaks in the clean dataset.

Input ATAC-seq datasets were divided into training, validation and holdout sets. The validation set consisted of data for chromosome 20 (for human data) or chromosome 11 (for mouse data), while the holdout set consisted of data for chromosome 10. Datasets for all other autosomes were included in the training set. These datasets were then further divided into non-overlapping intervals of 50 kb, unless otherwise specified (Supplementary Table 14), each representing a single training example. Each 50 kb long interval was padded with an additional 5 kb at either end, unless otherwise specified (Supplementary Table 14) so that the convolutional filter had enough neighboring bases to make predictions for every base inside the interval. Balanced training datasets with a fixed proportion of peaks were tested; however, this feature did not improve the overall performance metrics and was therefore not employed in genome-wide experiments.

**ResNet architecture used in AtacWorks**. The PyTorch neural network framework[36] was used to train a ResNet (residual neural network) model consisting of multiple stacked residual blocks. Each residual block included three convolutional layers and a skip layer to add the input to the first layer to the output of the third layer (Fig. 1a). Unless specified otherwise (Supplementary Table 14), each convolutional layer used 15 convolutional filters with a kernel size of 51 and a dilation of 8. Dilated convolutional layers were used to increase the receptive field of the model without increasing the parameter count. This approach has been effective in image classification tasks where a larger receptive field is desirable[37]. Models did not utilize batch normalization[38] for the convolutional layers, as it did not improve accuracy on either the regression or classification tasks in our experiments.

For each position in the given interval, the model performed two tasks; a regression or denoising task (predicting the ATAC-seq signal at each position) and a classification or peak calling task (predicting the likelihood that each position is part of a peak).

In order to perform both tasks, the input was passed through several residual blocks, followed by a regression output layer that returns the predicted ATAC-seq signal at each position in the input. The regression output was then passed through another series of residual blocks followed by a classification output layer that returned a prediction for whether each base in the input is part of a peak. The rectified linear unit (ReLU) activation function was used throughout the network, except for the classification output layer, which used a sigmoid activation function. The sigmoid activation forced the network to return a value between 0 and 1 for each input base, which was interpreted as the probability of that base being part of a peak. A cutoff of 0.5 was used to call peaks from these probability values.

Other convolutional neural network architectures, including the U-Net[39] were tested, and the selected architecture was chosen based on its robust performance in both denoising and peak calling tasks on several datasets.

**Model training in AtacWorks**. All deep learning models were trained using a multi-part loss function, comprising a weighted sum of three individual loss functions:

1. Mean squared error (MSE; for the regression output)
2. 1 - Pearson correlation coefficient (for the regression output)
3. Binary cross-entropy (for the classification output)

The relative importance of these loss functions was tuned by assigning different weights to each (Supplementary Table 14).

Training examples were randomly shuffled at the beginning of each training epoch and passed to the deep learning model in batches of 64 examples each, unless otherwise specified (Supplementary Table 14). At the end of each epoch of training, the performance of the model on the validation set was evaluated, and the model with the best validation set performance was saved and used.

Models were trained using the Adam optimizer[40] with a learning rate of $2 \times 10^{-4}$ for 25 epochs.

**Model evaluation in AtacWorks**. The performance of the model in regression was measured by computing Pearson correlation, Spearman correlation and MSE of the denoised data with respect to the clean dataset. For classification (peak calling), the model outputs the probability of belonging to a peak, for each position in the genome. In order to obtain predicted peaks, there is a set probability threshold above which a base is said to be a peak. Similarly, MACS2 produces a $p$ value for each position and the final peak calls depend on a user-defined probability threshold. Therefore the AUPRC and Area under the Receiver Operating Characteristic (AUROC) metrics were used to evaluate classification performance over the entire range of possible thresholds.

**Peak calling**. In order to call peaks from the base pair-resolution probabilities produced by AtacWorks, the macs2 bdgpeakcall command from MACS2 (v2.2.7)[14] was run with a threshold of 0.5. This is the same procedure used by MACS2 to call peaks from base pair-resolution $p$ values.

**Running AtacWorks**. AtacWorks v0.3.0 was used for all experiments.

All the parameters describing the models used in this paper are given in Supplementary Table 14. These parameters were chosen in a grid search based on validation set performance. Deeper and larger models produced slightly better results; however, larger models were also expensive and time-consuming to train.

AtacWorks took 2.7 min per epoch to train on one ATAC-seq dataset, and 22 min to test on a different whole genome, using 8 Tesla V100 16GB GPUs in an NVIDIA DGX-1 server.

**Paired high and low-quality ATAC-seq tracks**. Paired high and low-quality chromatin accessibility tracks were computationally generated from the same experiment in order to minimize the impact of potential batch effects. Published bulk ATAC-seq tracks from monocytes and erythroblasts[2] were split by technical and biological replicate, and then quantified using a TSS enrichment score. Tracks were then visually classified as high or low enrichment, and then aggregated based on classification and cell type to form the paired high and low-quality tracks (Supplementary Table 5). The original study describing these datasets found that ATAC-seq profiles were highly reproducible across both technical and biological replicates (mean Pearson $r = 0.94$ and $r = 0.91$, respectively)[2].

**Application of AtacWorks to dscATAC-seq of human blood**. Published dscATAC-seq datasets from human B cells, monocytes, and NK cells were obtained[17]. 2400 cells (~48 million reads) of each type were randomly selected to generate clean high coverage signal tracks and peak calls. Then, 1 cell (~20,000 reads), 5 cells (~100,000 reads), 10 cells (~200,000 reads), or 50 cells (~1 million reads) were randomly sampled the 2400 cells of each type to obtain noisy low-coverage data. For B cells and monocytes, this subsampling was repeated 5 times for 5, 10, and 50 cells, and 15 times for 1 cell in order to generate diverse training data. For each subsampling level, the data from B cells and monocytes was used to train a model, which was then tested on the corresponding subsampled data from NK cells. The 1 cell denoising model was tested on 4 different randomly chosen NK cells to obtain a more robust estimate of its performance.

**Application of AtacWorks to dscATAC-seq of mouse brain**. dscATAC-seq data from the mouse brain[18] was obtained, and 1800 cells (~48 million reads) each of the EN04 and EN12 excitatory neuron types were randomly selected to generate clean high coverage signal tracks and peak calls. Then, 4 cells (~100,000 reads), 8 cells (~200,000 reads) or 40 cells (~1 million reads) were randomly selected from among the 1800 cells of each type, to obtain noisy low-coverage data. The Atac-Works models trained on dscATAC-seq data from human B cells and monocytes were then applied to denoise these noisy datasets. Models were matched to low-coverage data based on sequencing depth; thus, the model trained on ten blood cells was applied to eight mouse brain cells as the latter had slightly higher sequencing depth. The denoised tracks and peak calls produced by AtacWorks were evaluated by comparing them to the clean tracks and peak calls for the same cell types.

**Application of AtacWorks to sciATAC-seq and dsciATAC-seq**. Two experiments were performed to test whether AtacWorks models could generalize to single-cell data sequenced using different platforms.

First, dsciATAC-seq data from human blood cell types was obtained[17]. Data was aggregated over 20,378 CD4+ T cells to generate a clean high coverage signal track (~43 million reads) and peak calls. 450 cells (~1 million reads) were subsampled to obtain noisy low-coverage data. The dscATAC-seq model trained on 50 human blood cells (described above) was applied to denoise and call peaks from this noisy dataset. The denoised tracks and peak calls produced by AtacWorks were evaluated by comparing them to the clean high coverage signal track. Due to the low sequencing depth for other cell types in this dataset, it was not possible to generate sufficiently high-coverage clean tracks for any other cell types besides CD4+ T cells.

Second, sciATAC-seq data from a mouse lung tumor was obtained[18]. The sequencing depth in this dataset was insufficient to obtain clean datasets of >40

million reads as described in the previous examples. Instead, clean coverage tracks and peak calls were obtained by aggregating data over 550 cells (~15 million reads) each of B cells, monocytes and macrophages. 35 cells (~1 million reads) were randomly sampled from among the 550 cells of each type to obtain noisy low-coverage data. For B cells and monocytes, this subsampling was repeated five times in order to generate diverse training data. The data from B cells and monocytes was used to train a model, which was then tested on the subsampled data from the macrophages.

To demonstrate the feasibility of cross-platform application, dscATAC-seq data from human monocytes and B cells was subsampled to generate data of the same sequencing depths as the sci-ATAC data, i.e., 700 cells (~15 million reads) were aggregated to generate clean data, and 50 cells (~1 million reads) were aggregated to generate noisy data. Subsampling was repeated five times to generate diverse training data. A model was trained using this dscATAC-seq data and was applied to the subsampled sciATAC-seq dataset of macrophages.

**Adding binding motif locations for CTCF ChIP-seq prediction.** The deep learning model was modified to take additional inputs along with the noisy ATAC-seq signal. Potential CTCF (CCCTC-binding factor) binding sites were identified on both strands of the genome using motifmatchr (https://github.com/GreenleafLab/motifmatchr). The top 200,000 sites were selected and expanded to 500 bp regions centered on the known binding motif. In order to predict CTCF ChIP-seq peaks from ATAC-seq data, the model was given the positions of CTCF binding motifs on the genome in addition to the noisy ATAC-seq coverage track. For every position in the genome, the model received three numeric inputs: the coverage at that position in the noisy ATAC-seq dataset, a 0 or 1 representing whether that position was part of a CTCF binding motif on the forward strand, and a 0 or 1 representing whether that position was part of a CTCF binding motif on the reverse strand.

**Generation of dscATAC-seq data for FACS-isolated HSCs.** Cryopreserved human BMMCs were purchased from Allcells (catalog number BM, CR, MNC, 10 M). Cells were quickly thawed in a 37 °C water bath, rinsed with culture medium (RPMI 1640 medium supplemented with 15% FBS) and then treated with 0.2 U/µL DNase I (Thermo Fisher Scientific) in 2 mL of culture medium at room temperature for 15 min. After DNase I treatment, cells were filtered with a 40 µm cell strainer, washed with MACS buffer (1x PBS, 2 mM EDTA and 0.5% BSA), and cell viability and concentration were measured with trypan blue on the TC20 Automated Cell Counter (Bio-Rad). Cell viability was greater than 90% for all samples. CD34$^+$ cells were then bead enriched using the CD34 MicroBead Kit UltraPure (Miltenyi Biotec, catalog number 130-100-453) following manufacturer's instructions. The enriched population was then simultaneously stained with CD45, Lineage cocktail, CD34, CD38, CD45RA and CD90 antibodies in MACS buffer for 20 min at 4 °C, using the following antibody dilutions: CD45 (BV711; BioLegend #304050) - 1:100, Lineage cocktail (FITC; BioLegend #348801) - 1:25, CD34 (APC-Cy7; BioLegend #343514) - 1:50, CD38 (PE-Cy7; BioLegend #303516) - 1:50, CD45RA (BUV737; BD Biosciences #612846) - 1:50, CD90 (BV421; BioLegend #328122) - 1:25. Stained cells were then washed with MACS buffer and the CD45$^+$ Lin$^-$ CD38$^-$ CD34$^+$ CD45RA$^-$ CD90$^+$ fraction (HSCs) was FACS sorted using a MoFlo Astrios EQ Cell Sorter (Beckman Coulter), using the Beckman Coulter MoFlo Astrios EQ Cell Sorter's Summit v62 software to collect the data. The FACS data was analyzed using FlowJo v10.7, and the gating strategy is shown in Supplementary Fig. 9.

Single-cell ATAC-seq data was then generated for the sorted HSCs using the dscATAC-seq Whole Cell protocol as described in Lareau et al.[17]. For a detailed description of tagmentation protocols and buffer formulations, refer to the SureCell ATAC-Seq Library Prep Kit (17004620, Bio-Rad) User Guide (10000106678, Bio-Rad). Briefly, the sorted HSCs were resuspended in Whole Cell Tagmentation Mix containing 0.1% Tween-20, 0.01% digitonin, 1× PBS supplemented with 0.1% BSA, ATAC Tagmentation Buffer and ATAC Tagmentation Enzyme. Cells were mixed and agitated on a ThermoMixer (5382000023, Eppendorf) for 30 min at 37 °C. Tagmented cells were kept on ice before being loaded onto a ddSEQ Single-Cell Isolator (12004336, Bio-Rad). scATAC-seq libraries were prepared using the SureCell ATAC-Seq Library Prep Kit (17004620, Bio-Rad) and SureCell ATAC-Seq Index Kit (12009360, Bio-Rad). Bead barcoding and sample indexing were performed in a C1000 Touch thermal cycler with a 96-Deep Well Reaction Module (1851197, Bio-Rad); PCR conditions are as follows: 37 °C for 30 min, 85 °C for 10 min, 72 °C for 5 min, 98 °C for 30 s, eight cycles of 98 °C for 10 s, 55 °C for 30 s and 72 °C for 60 s, and a single 72 °C extension for 5 min to finish. Emulsions were broken and products were cleaned up using Ampure XP beads (A63880, Beckman Coulter). Barcoded amplicons were further amplified using a C1000 Touch thermal cycler with a 96-Deep Well Reaction Module; PCR conditions are as follows: 98 °C for 30 s, seven cycles of 98 °C for 10 s, 55 °C for 30 s and 72 °C for 60 s, and a single 72 °C extension for 5 min to finish. PCR products were purified using Ampure XP beads and quantified on an Agilent Bioanalyzer (G2939BA, Agilent) using the High-Sensitivity DNA kit (5067-4626, Agilent). Libraries were loaded at 1.5 pM on a NextSeq 550 (SY-415-1002, Illumina) using the NextSeq High Output Kit (150 cycles; 20024907, Illumina) and sequencing was performed using the following read protocol: read 1, 118 cycles; i7 index read, 8 cycles; read 2, 40 cycles.

A custom sequencing primer (part of the SureCell ATAC-Seq Library Prep Kit) is required for read 1.

**Preprocessing of dscATAC-seq data for FACS-isolated HSCs.** Per-read bead barcodes were parsed and trimmed using UMI-tools v1.0.0[41]. Constitutive elements of the bead barcodes were assigned to the closest known sequence allowing for up to 1 mismatch per 6-mer or 7-mer (mean >99% parsing efficiency across experiments). Paired-end reads were aligned to hg19 using BWA v0.7.17[42] on the Illumina BaseSpace online application. Bead-based ATAC-seq processing (BAP)[17] was used to identify systematic biases (i.e., reads aligning to an inordinately large number of barcodes) and barcode-aware deduplicate reads, as well as perform merging of multiple bead barcode instances associated with the same cell. Barcode merging was necessary due to the nature of the BioRad SureCell scATAC-seq procedure used in this study, which enables multiple beads per droplet. BAP was given an alignment (.bam) file for a given experiment with a bead barcode identifier indicated by a SAM tag as input. Aligned reads were combined using samtools merge (v1.9).

**Bulk-guided projection of single cells.** The bulk-guided UMAP projection of single cells (Fig. 2c) was performed as described in Lareau et al.[17]. In brief, a common set of peaks ($k = 156,311$) was used to create a vector of read counts for each CD34$^+$ single-cell ATAC-seq profile. Principal Component Analysis (PCA) was run on published bulk ATAC-seq data[2] to generate eigenvectors capturing variations in regulatory element accessibility across cell types. Each single cell was then projected in the same space as these eigenvectors by multiplying their counts vector by the common PCA loading coefficients. The resulting projection scores were scaled and centered prior to being visualized using UMAP. Predicted labels for the CD34$^+$ cells were derived by correlating their projected single-cell scores with those of a reference set of FACS-isolated PBMCs[24] and assigning the label of the closest match.

**Transcription factor motif accessibility z-scores.** Motif accessibility z-scores for GATA2 (Fig. 2d) were computed using chromVAR version 1.12.0[27]. The method calculates enrichment or depletion in accessibility within peaks that share a common transcription factor motif while adjusting for GC content and overall region accessibility. The single cells were scored using a list of human transcription factor motifs from the CIS-BP database (http://cisbp.ccbr.utoronto.ca/index.php).

**Smoothed gene accessibility scores.** Gene accessibility scores for MEF2C (Fig. 2e) were computed as described in Lareau et al.[17]. Briefly, to obtain gene scores for a particular gene across all cells, any sequencing reads within 10 kb of the gene's transcription start site were compiled and weighted using an inverse exponential decay function. The weighted reads were then summed for each cell and smoothed by averaging the scores from each cell's 50 nearest neighbors in principal component space. A list of TSSs for hg19 was obtained from the UCSC Table Browser (https://genome.ucsc.edu/cgi-bin/hgTables).

**Denoising lineage-priming HSCs with AtacWorks.** Each subsample of lineage-priming HSCs was generated by selecting a single HSC and aggregating the 50 most similar HSCs in principal component space. The selected HSCs were chosen and annotated based on their proximity to specific populations of labeled CD34$^+$ cells (Fig. 2b). After aggregation, three resulting subsamples were converted from BAM to bigWig format as described in Data Preprocessing and denoised using a model trained on dscATAC-seq data from B cells and monocytes. The denoised tracks were then normalized by coverage for cross-sample comparisons.

**Denoising of randomly permuted HSCs with AtacWorks.** A list of differentially expressed genes in blood cells was obtained from the Human Cell Atlas Data Portal (https://data.humancellatlas.org) and filtered down to a set of 2303 genes relevant to HSCs. Transcription start sites (TSSs) for each of these genes were obtained and expanded by 100 kb in both directions to generate a set of hematopoiesis regulatory regions comprising around 300 million bases, or 10% of the genome.

To provide a background model for the denoised lineage-primed samples, 1000 subsamples were generated by randomly selecting 1000 HSCs from the pool of 9974 and for each selected cell, aggregating the 50 most similar HSCs in principal component space. These random samples were converted from BAM to bigWig format as described in Data Preprocessing. The 1000 random samples were then denoised using the same AtacWorks model used to denoise the lineage-priming HSCs, but only in defined hematopoiesis regulatory regions, reducing the runtime by over 90%. The denoised random samples were normalized by coverage. For each genomic position in the hematopoiesis regulatory regions, a mean and standard deviation of coverage was calculated across the 1000 denoised random samples.

For each subsample of lineage-primed HSCs, a z-score for each genomic position in the hematopoiesis regulatory regions was generated based on the normalized coverage relative to the mean and standard deviation in the 1000 denoised random samples. Regulatory peaks were called by combining all genomic positions with an absolute z-score >2 within 200 bp of each other. The top z-scores for each peak were converted to p values and then corrected for multiple hypothesis

testing using the Benjamini Hochberg procedure. All peaks with a false discovery rate <0.05 were saved in a BED file. Peaks were then filtered by a minimum coverage value to remove low-coverage regions that would not be identified through typical ATAC-seq analysis. BED files were converted to bigWig format for visualization using the bedGraphtoBigWig utility (v4).

**Validation of putative regulatory elements.** The set of 28,505 bead-enriched CD34+ bone marrow progenitor cells was loaded into chromVAR[27] as described in Lareau et al.[17]. In brief, the published BAM files were converted to a cells by peaks matrix, where each matrix element represents the sequencing coverage. The sets of filtered genomic regions for each subsample of lineage-primed HSCs were then loaded as discrete annotations. The chromVAR computeDeviations function was then used to quantify the normalized accessibility of each of these subsets in every CD34+. The resulting accessibility z-scores were then visualized on the UMAP projection to confirm that the identified lineage-priming elements were generally more accessible in the corresponding differentiated cell populations. To quantify the most differentially-accessible transcription factor motifs across these elements, a new counts matrix was generated in chromVAR, but only from HSCs and the identified lineage-priming peaks. The overlap between the peaks and transcription factor motifs was found, and then the normalized accessibility of any overlapping motifs was calculated using the computeDeviations function. Lastly, the variability of each motif was calculated using the computeVariability function.

**Data visualization.** Unless otherwise specified, the WashU epigenome browser (http://epigenomegateway.wustl.edu/browser/) was used for ATAC-seq signal track visualization. The denoised lineage-priming HSC subsamples (Fig. 3f) were visualized using the Integrative Genomics Viewer[43].

**Reporting summary.** Further information on experimental design is available in the Nature Research Reporting Summary linked to this paper.

## Data availability
Bulk ATAC-seq datasets of various blood cell types are available from GEO under accession number "GSE74912 [https://www.ncbi.nlm.nih.gov/geo/query/acc.cgi?acc=GSE74912]". From these datasets, B cells, NK cells, CD4+ and CD8+ T cells were used for model training, while erythroblasts and monocytes were used for testing. For the transcription factor footprinting model, NK cells, CD8+ T cells, and multipotent progenitor (MPP) cells were used for training, while HSCs were used for testing. The bulk ATAC-seq dataset for Peyer's Patch is available from ENCODE under experiment "ENCSR017RQC [https://www.encodeproject.org/experiments/ENCSR017RQC/]".

The dscATAC-seq dataset of hematopoietic stem cells generated for this study is available from GEO under accession number "GSE147113 [https://www.ncbi.nlm.nih.gov/geo/query/acc.cgi?acc=GSE147113]".

Other dscATAC-seq and dsciATAC-seq datasets are available from GEO under accession number "GSE123581 [https://www.ncbi.nlm.nih.gov/geo/query/acc.cgi?acc=GSE123581]". From these datasets, CD4+ T cells, CD8+ T cells, and pre-B cells were used for model training, while monocytes were used for testing. Bead-isolated CD34+ cells were used for the combined UMAP projection. The sciATAC-seq datasets of B cells, monocytes, and macrophages from primary lung tumor are available from GEO under accession number "GSE145194 [https://www.ncbi.nlm.nih.gov/geo/query/acc.cgi?acc=GSE145194]". B cells and monocytes were used for model training, while macrophages were used for testing. The scATAC-seq dataset of FACS-isolated peripheral blood mononuclear cells (PBMCs) is available from GEO under accession number "GSE96772 [https://www.ncbi.nlm.nih.gov/geo/query/acc.cgi?acc=GSE96772]". These cells were used to infer cell type labels for CD34+ cells in the combined UMAP projection.

CTCF ChIP-seq tracks are available from ENCODE under experiments "ENCSR000DLK [https://www.encodeproject.org/experiments/ENCSR000DLK/]" (HSCs), "ENCSR000ATN [https://www.encodeproject.org/experiments/ENCSR000ATN/]" (Monocytes) and "ENCSR000AUV [https://www.encodeproject.org/experiments/ENCSR000AUV/]" (B cells). H3K27ac ChIP-seq tracks are available from ENCODE under experiments "ENCSR000AUP [https://www.encodeproject.org/experiments/ENCSR000AUP/]" (B cells) and "ENCSR000ASJ [https://www.encodeproject.org/experiments/ENCSR000ASJ/]" (monocytes).

The list of human transcription factor motifs was curated from the CIS-BP database (http://cisbp.ccbr.utoronto.ca/index.php) and is available at https://github.com/GreenleafLab/chromVARmotifs. The list of transcription start sites for hg19 was obtained from UCSC Table Browser (https://genome.ucsc.edu/cgi-bin/hgTables). The list of differentially expressed genes in blood cells was curated from the Human Cell Atlas Data Portal (https://data.humancellatlas.org) and is available at https://github.com/zchiang/atacworks_analysis.

All of the processed data, trained models, and output signal tracks described in this paper are publicly available at https://atacworks-paper.s3.us-east-2.amazonaws.com.

All other relevant data supporting the key findings of this study are available within the article and its Supplementary Information files or from the corresponding author upon reasonable request. Source data are provided with this paper. A reporting summary for this Article is available as a Supplementary Information file. Source data are provided with this paper.

## Code availability
AtacWorks is available at https://github.com/clara-parabricks/AtacWorks[12]. Custom scripts used to batch process samples for input and identify differentially-accessible regulatory regions in lineage-primed hematopoietic stem cells are available at https://github.com/zchiang/atacworks_analysis[44].

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

## Acknowledgements

We thank Eric Xu, Joyjit Daw, Neha Tadimeti and Ohad Mosafi for contributing to the code for AtacWorks. We thank Ronald Lebofsky and Giulia Schiroli for assistance in generating dscATAC-seq data. We thank Yan Hu for critical reading of the paper. We thank members of the Buenrostro lab and NVIDIA team for insightful comments throughout the development of this work. J.D.B., Z.D.C., and F.M.D. acknowledge support by the Allen Distinguished Investigator Program through the Paul G. Allen Frontiers Group. This work was further supported by the Chan Zuckerberg Initiative and the NIH Director's New Innovator award. Z.D.C. is supported by the NSF-Simons Center for Mathematical and Statistical Analysis of Biology at Harvard (#1764269).

## Author contributions

N.Y. and A.L. developed the deep learning model. A.L. and Z.D.C. performed data analysis. F.M.D. performed HSC dscATAC-seq experiments. A.L., Z.D.C., and J.D.B. wrote the paper with input from all authors. J.I. and J.D.B. jointly conceptualized and supervised this work. A.L. and Z.D.C. contributed equally.

## Competing interests

J.D.B. holds patents related to ATAC-seq and is a member of the scientific advisory board for Camp4, Seqwell, and Celsee. A.L., N.Y., and J.I. are employees of NVIDIA Corporation. All other authors declare no competing interests.
