## [Peer Review File · Nature Communications]

REVIEWER COMMENTS

Reviewer #1 (Remarks to the Author):

The authors present AtacWorks as a deep NN that trains on bulk ATAC data and is used to denoise low depth or single-cell ATAC-seq data. I believe that the framework has potential; however, the applications shown are limited (only on blood cell types) and only on one single-cell ATAC-seq platform that produces notably high quality data. It is also unclear how similar the training set data needs to be to use the tool or if pure cell type bulk ATAC-seq data is required. If that is the case it will be very limited in its applications, as single-cell assays are most useful when pure cell types can not be physically separated out for bulk assays.

Line 106 states that AtacWorks requires high coverage data from the same cell type as single-cell; when often a bulk ATAC library contains many cell types – hence the need for single-cell. It seems that if there is no challenge to isolate cell types (as in many hematopoietic cell types) then there is less need for single-cell.

I appreciate the power to call peaks from low-coverage or pseudobulk single-cell data of low cell number; however the requirement to have matching bulk ATAC-seq data seems to be an issue. I acknowledge that these pairings of bulk and single-cell are used to train the model and then the model can be applied to other cell types where bulk is not possible, but how similar does the training data have to be?

The authors claim diverse applications; however they only focus on blood cell data. I would like to see it applied to other datasets and cell types, such as the dscATAC-seq mouse brain dataset as well as lower-quality single-cell ATAC-seq data like from the original snATAC-seq work from the Ren Lab (Preissl et al) or on the cusanovich et al 2018 Cell paper. To make a claim of generalizability, it should be demonstrated on a variety of data sets and methods.

I am somewhat skeptical of the cross-modality predictions. CTCF has a well-known nucleosome positioning signal for +/- 5 or so nucleosomes that could drive this inference – it also has a very distinct and large binding motif and is also one of the most readily footprinted factors. Then H3K27ac overlaps very closely to ATAC data - one would need to identify sites that are open and peaks in ATAC that lack H3K27ac and show it does not predict binding there – as the authors will be hard-pressed to find a site that is confidently closed chromatin yet has H3K27ac marks.

Reviewer #2 (Remarks to the Author):

Lal and colleagues present AtacWorks, a deep learning method to make up for the limitations of low quality or low coverage ATAC-seq datasets, with an immediate application for single cell datasets. The manuscript is overall very clear and the different analyses show that AtacWorks seems a promising tool to overcome the limitations of epigenomic datasets. Here are some comments and questions which need clarification and some analysis to further validate and test the potential of AtacWorks.

1/ The authors compare the performance of AtacWorks to one existing convolutional network approach from Rai et al. Additional benchmarking with other recently published deep convolutional network approach for single cell datasets, such as SCALE and scVI – which perform denoising and imputation- should be performed to compare performance, (even if these models don't necessarily take as input the same types of data).

2/ Regarding false positive, for example in Fig. 1f, the reader appreciates by looking at the tracks the denoising of the low-quality datasets by AtacWorks; yet a series of signal bumps appear on the track as well, suggesting that there could be false positive signal on the denoised coverage track. While it does seem to affect the AUPRC/AUROC metrics, which are based on the peak called by AtacWork, it could be an issue if the coverage track is used for other purposes than peak calling. The authors should further study and comment this issue of the coverage track, and potentially

propose solutions to remove such signal.

3/ Concerning the validation of the denoising procedure, for example in the case of Fig. 2f, with single-cell cumulative bulks, peaks are discovered by AtacWorks in sub-populations. How could these be experimentally or in silico validated to check that these are not false positives?

4/ In order to verify that the model is not simply learning features from the training set, the authors show that regions not in the training set (chromosome 10) are successfully denoised from the low-coverage experiment (Fig 1c, d, Supplementary Table 1). Additionally, they show that the results were highly robust to different subsets, when using a series of hematopoietic cells as training datasets (Supplementary Table 3). While these results are a good evidence that the model is not overfitting the training set, it would be very interesting to question more deeply the impact of the training dataset. For instance, we would recommend comparing the results obtained for one test set with a series of very different cell types. Indeed, the vast majority of datasets originate from hematopoietic cells, the authors should add ATAC-seq from a couple of different tissues to challenge the effect of the training set on the performance of the model.

5/We had several remarks about the usability of AtacWork for single-cell datasets. Could AtacWork be used for peak inference only on individual cells for example? Why do the authors limit their usage of AtacWork on groups of 50 cells precisely? Overall we could expect more insight on the application of AtacWork to scATAC-seq.

Reviewer #3 (Remarks to the Author):

This manuscript presents AtacWorks, a tool for denoising low-coverage and low-quality ATAC data. AtacWorks looks like a well-written piece of software. Furthermore, the paper presents convincing demonstrations of its utility. AtacWorks is an excellent contribution to the community and I believe it will get a lot of use. The paper is well-organized and well-written I have no problem recommending publication of the paper as is.

I have only a few minor suggestions and comments that I offer for your consideration:

1. I believe you never actually present results of denoising a single cell individually -- always aggregated into small groups (of 50). Presumably, this is because the single cell is simply too little data for the method. Can you comment on this?

2. It's interesting that AtacWorks can pick up signals not found in the training data. However, this would have been more convincing if you'd have used a non-blood sample for your test case. As is, you trained the model using a variety of different blood cell types, and then your held-out sample was another blood cell type. It would be interesting if this capacity is reduced the further away you get from the training data. But perhaps more important, I would like to know if you can tease out of the network any information about exactly what features contribute to this ability to predict outside of the training data. I know deep networks are notorious for difficulty of interpretation, but nevertheless, perhaps some feature importance approaches may aid in interpreting the model.

3. For the peak calling component of the model, you state that the classifier operates on each nucleotide individually. How do you prevent the model from learning discontinuous peaks, which are not realistic? Considering each nucleotide independently seems likely to lead to such a scenario, where the peaks are spotty throughout the genome.

4. For the final analysis with 1000 samples of 50 randomly selected HSCs, it's not clear to me that this is the best background distribution to draw from, as I would expect the denoised tracks to be less useful with 50 random cells than with 50 nearest neighbors to a single cell. It seems a more appropriate analysis would be to select 1000 random single cells, and then choose the 50 nearest neighbors to each, and then use that to build a null distribution.

5. The github URLs appear to be outdated.

Reviewer #4 (Remarks to the Author):

This paper focuses on improving the signal resolution of ATAC-seq data of chromatin accessibility at single-cell level. The authors introduce a deep learning based model, named AtacWorks, to analyze ATAC-seq data so that biologically-meaningful change could be measured more efficiently. Based on Resnet, AtacWorks integrates three types of optimized function, two for data denoising and one for peak calling. The model was applied to multiple tasks, including denoising single-cell aggregated data, cross-modality prediction, and enhancing the resolution of rare subpopulations of single cells. These results are all validated across different cell types demonstrating the model robustness.

This manuscript is well-written. For further improvement the authors could address the following comments.

Major comments

1. While the main contribution lies in the development of a new method, the description of the method looks a bit superficial. Readers can see that AtacWorks has good performance, but there is little insight into why. In particular, why is Resnet more suitable than the U-Net previously used for the tasks here? If the authors of the U-Net method do some parameter tuning, could it outperform AtacWorks in turn? The results comparison with U-Net should be shown more clearly in the results section.
2. The cross-modality predictions look interesting. However, what is the motivation for doing that, considering that scATAC-seq is probably no more reliable than the data of other modalities? Maybe ATAC-seq could be used for interpreting or complementing other data, or the authors may just want to show the reliability of AtacWorks. Please explain it after line 218.
3. From the perspective of learned patterns or model performance, please explain the model's generalization on unseen cell types.

Minor Comments

1. The training datasets had better include at least one collected from real experiments rather than only subsampling from high-converge dataset.
2. The runtime comparison with U-Net is unfair, since U-Net was implemented on CPU while AtacWorks is on GPU. For a fair comparison, the U-Net-based model should also be trained on GPU with the same environment as AtacWorks.
3. For "potential for AtacWorks to be broadly applied for cross-modality inference of latent epigenetic states", the authors could have more analysis on either model training or results to demonstrate this potential.
4. The full spelling of "HSCs" should be given when the word firstly appears in the article.
5. As stated in the paper, the so-called "potential regulatory elements" have not been validated, therefore could only be called "novel signals" since they may also be artifacts due to some bias introduced by the predictive model. Some validation or analysis should be given here.

Reviewer #1

Comment 1.1: The authors present AtacWorks as a deep NN that trains on bulk ATAC data and is used to denoise low depth or single-cell ATAC-seq data.

Response 1.1: We thank the reviewer for their helpful comments throughout our manuscript. We address each of their comments below, and have colored significant revisions to the main text dark blue for clarity.

We would like to clarify that AtacWorks can be trained on either bulk or single-cell ATAC-seq data. In fact, the models that we applied to denoise and call peaks from single-cell data were also trained on single-cell data. We acknowledge that this was not clear enough in our original text. To address this concern, we have added a new schematic in Fig. 2a illustrating how AtacWorks can be trained on single-cell data, copied below:

Fig. 2a

Schematic showing the strategy to train and test AtacWorks on single-cell ATAC-seq data. A clean high-coverage ATAC-seq signal is obtained by aggregating data from all cells belonging to an abundant cell type. Data is aggregated over a randomly selected subset of these cells to obtain a noisy signal. Paired clean and noisy datasets are used to train an AtacWorks model. The model can be applied to denoise and call peaks from noisy aggregate data from small numbers of cells, either from the same experiment or a different experiment.

We have also rewritten sections of the main text, copied below:

To demonstrate our method is also adaptable to broad use cases of ATAC-seq, we applied AtacWorks to denoise data from a high-throughput single-cell ATAC-seq experiment. We first obtained droplet single-cell ATAC-seq (dscATAC-seq) data from bead-isolated human blood cells and aggregated single-cell chromatin accessibility profiles by cell type¹⁶. We selected two cell types (B cells and monocytes) from the dataset, and produced clean ATAC-seq signal tracks and peak calls by aggregating data over 2400 cells (~50 million reads) of each type. We then generated noisy ATAC-seq signals by randomly subsampling subsets of cells of each type, and trained AtacWorks models on the paired clean and noisy datasets (Fig. 2a). We randomly sampled 1 cell (~20,000 reads), 5 cells (~0.1 million reads), 10 cells (~0.2 million reads) or 50 cells (~1 million reads) for the low-coverage training datasets. The resulting trained models improved signal track accuracy and peak calling from aggregated natural killer (NK) cells sequenced using the same protocol (Fig. 2b, Supplementary Table 8, Supplementary Table 9). Notably, AtacWorks improved the AUPRC of peak calls from 50 NK cells from 0.2048 to 0.7008, a result that MACS2 requires over 400 cells to obtain (Fig. 2b, Supplementary Table 8).

Comment 1.2: I believe that the framework has potential; however, the applications shown are limited (only on blood cell types) and only on one single-cell ATAC-seq platform that produces notably high quality data.

Response 1.2: To address the reviewer's concern of limited applications, we have applied AtacWorks models trained on droplet single-cell ATAC-seq (dscATAC-seq) of sorted human blood cells (B cells and monocytes) to:

- 1) dscATAC-seq data from mouse brain (Lareau et al. 2019) to show generalizability across diverse cell types and species (Figure 2c and 2d, Supplementary Table 10)
- 2) Single-cell data from human blood cells, sequenced using the dsciATAC-seq platform (Lareau et al. 2019) to show generalizability across platforms and data quality (Supplementary Table 11).
- 3) Single-cell data from macrophages from mouse primary lung tumors, sequenced using the sci-ATAC-seq platform (LaFave et al. 2020) to show generalizability across species, platforms and data quality (Supplementary Table 12).

We have revised the main text to reflect these new analyses, copied below:

We then tested whether these models trained on dscATAC data from human blood could generalize to non-blood cell types. To do this, we obtained single-cell data from a mouse brain using the same dscATAC protocol¹⁷. We then applied the models trained on human blood to data aggregated from mouse pyramidal and oscillatory neurons. For both types of neurons, we observed that AtacWorks improved the signal track and peak calls, both overall and within cell-type specific peaks (Fig. 2c, d, Supplementary Table 10). This result demonstrates that AtacWorks is broadly applicable across both cell types and species.

Finally, because the previous experiment was limited to dscATAC data, we sought to investigate the generalizability of AtacWorks models to data from different single-cell platforms. To this end, we applied one of the previously-described AtacWorks models trained on dscATAC-seq data to human CD4⁺ T cells sequenced using a combinatorial indexing-based platform (dsciATAC-seq¹⁷), and observed improvements in both the signal track and peak calls (Supplementary Table 11). We also applied a similar model trained on dscATAC-seq data from human blood to macrophages from mouse primary lung tumors sequenced using

the sciATAC-seq protocol¹⁸. Once again, we observed that the model trained on human dscATAC-seq data improved both signal track accuracy and peak calls. (Supplementary Table 12). However, we note that a model trained on sciATAC-seq data from B cells and monocytes returned slightly better results on most metrics when applied to the same sciATAC-seq dataset from macrophages (Supplementary Table 12). Collectively, these results support AtacWorks as a highly generalizable tool to study single-cell ATAC-seq data.

We note that although the dscATAC-seq and dsciATAC-seq technologies share an underlying droplet-based platform, the dsciATAC-seq data is of notably lower quality, likely due to the combinatorial barcoding step unique to that method. However, due to their commonalities, we have also added sci-ATAC-seq data to demonstrate application to a platform that uses exclusively combinatorial barcoding.

Comment 1.3: It is also unclear how similar the training set data needs to be to use the tool or if pure cell type bulk ATAC-seq data is required. If that is the case it will be very limited in its applications, as single-cell assays are most useful when pure cell types can not be physically separated out for bulk assays.

Line 106 states that AtacWorks requires high coverage data from the same cell type as single-cell; when often a bulk ATAC library contains many cell types – hence the need for single-cell. It seems that if there is no challenge to isolate cell types (as in many hematopoietic cell types) then there is less need for single-cell. I appreciate the power to call peaks from low-coverage or pseudobulk single-cell data of low cell number; however the requirement to have matching bulk ATAC-seq data seems to be an issue.

Response 1.3: We acknowledge that the requirements for the training data were unclear. To address this concern, we have rewritten the main text to reflect that bulk ATAC-seq from a pure cell type is not required for training. AtacWorks models can be trained on many different types of data, including bulk ATAC-seq data from a pure cell type, bulk ATAC-seq data from mixed cell types, or single-cell ATAC-seq data.

To demonstrate this, we have trained a model on ENCODE bulk ATAC-seq data from three human tissues (coronary artery, tibial nerve, and left ventricle) containing a mixture of cell types. We applied this model, as well as a model trained on data from FACS-isolated cell types, to denoise bulk ATAC-seq data from the human Peyer's patch (also containing mixed cell types). We have revised the main text to include this analysis (copied below) and the results are presented in Supplementary Fig. 3 and Supplementary Table 4.

Since ATAC-seq is commonly applied to tissues containing a mixture of cell types, we sought to test whether our models could be applied to data of this nature. We found that a model trained on FACS-isolated cell types from human blood was able to denoise subsampled low-coverage ATAC-seq data from a mixture of human cell types derived from the intestinal Peyer's Patch by the ENCODE consortium^{14,15} (Supplementary Fig. 3, Supplementary Table 4). This suggests that our models are robust to data from mixtures of cell types, as well as varied experimental preparation of cells and tissues. However, we note that a model trained on three different ENCODE datasets produces better results on this task (Supplementary Table 4), suggesting that results may be improved when the training and test data are obtained from the same experimental protocol.

To address the reviewer's concerns about limited use cases on single-cell data, we have trained models using single-cell data and applied them to a wide variety of single-cell datasets from different cell types, species, and platforms to highlight the diverse applications of AtacWorks (see Responses 1.1 and 1.2).

Comment 1.4: I acknowledge that these pairings of bulk and single-cell are used to train the model and then the model can be applied to other cell types where bulk is not possible, but how similar does the training data have to be?

Response 1.4: We acknowledge that the generalizability of AtacWorks models was not fully explored in our original manuscript. To address this concern, we applied an AtacWorks model trained on dscATAC-seq of human blood cells to several single-cell datasets from different cell types, species, and platforms (see Response 1.2). Overall, we observed that while our models are highly generalizable, there are small benefits to matching training and test data when possible. We have added this point to our Discussion section:

In addition to generalization across different cell types, we also observed that our trained models can generalize to data from different species, experimental platforms, and quality levels. However, we observed that we could obtain slightly better results (e.g. AUPRC increase from 0.7332 to 0.7483) on a test dataset by using a model trained on more closely matched data (Supplementary Table 12), suggesting that there remain small benefits to matching training and test data when possible.

Comment 1.5: The authors claim diverse applications; however they only focus on blood cell data. I would like to see it applied to other datasets and cell types, such as the dscATAC-seq mouse brain dataset as well as lower-quality single-cell ATAC-seq data like from the original snATAC-seq work from the Ren Lab (Preissl et al) or on the cusanovich et al 2018 Cell paper. To make a claim of generalizability, it should be demonstrated on a variety of data sets and methods.

Response 1.5: We agree with the reviewer that demonstrations on a variety of data sets and methods are required to claim generalizability. As suggested, we have trained a model on single-cell (dscATAC-seq) data from human blood and applied it to dscATAC-seq of mouse brain to show cross-cell type and cross-species generalizability. We have also used the same model to denoise “lower-quality” (in terms of reads per cell) sci-ATAC-seq data of normal cells from mouse primary lung tumors. This data was generated using a similar combinatorial barcoding approach as was used to generate the two datasets suggested by the reviewer, and support our claim of cross-platform generalizability. See Response 1.2 for more details.

Comment 1.6: I am somewhat skeptical of the cross-modality predictions. CTCF has a well-known nucleosome positioning signal for +/- 5 or so nucleosomes that could drive this inference – it also has a very distinct and large binding motif and is also one of the most readily footprinted factors. Then H3K27ac overlaps very closely to ATAC data - one would need to identify sites that are open and peaks in ATAC that lack H3K27ac and show it does not predict binding there – as the authors will be hard-pressed to find a site that is confidently closed chromatin yet has H3K27ac marks.

Response 1.6: We agree with the reviewer’s observations about CTCF and H3K27ac. Though we remain excited about the potential of AtacWorks to broadly perform cross-modality predictions, we have revised portions of the main text to more accurately reflect the limitations of the current approach, copied below:

These cross-modality predictions demonstrate the potential for AtacWorks to generate multiple layers of information in single cells from one of the most commonly-used epigenomic assays, at no additional cost. It is generally experimentally challenging to make multiple measurements from the same cells, so this approach may be especially useful in cases where running multiple ChIP-seq experiments is infeasible due to time, reagents, sample availability, or biological variability. Though the models presented here tend to perform better on active histone marks (e.g. H3K27ac) or abundant architectural proteins (e.g. CTCF), these specific predictions may be useful for distinguishing active vs. poised enhancers²¹ or characterizing changes in 3D genome structure across differentiation²². We anticipate future work will extend these capabilities to enable cross-modality inference of additional latent epigenetic states from a single experiment.

Reviewer #2

Comment 2.1: Lal and colleagues present AtacWorks, a deep learning method to make up for the limitations of low quality or low coverage ATAC-seq datasets, with an immediate application for single cell datasets. The manuscript is overall very clear and the different analyses show that AtacWorks seems a promising tool to overcome the limitations of epigenomic datasets. Here are some comments and questions which need clarification and some analysis to further validate and test the potential of AtacWorks.

Response 2.1: We thank the reviewer for their comment that AtacWorks is a “promising tool” and hope that our revised manuscript provides sufficient clarification for their questions. We address the rest of their comments below, and have colored significant revisions to the manuscript dark blue for clarity.

Comment 2.2: The authors compare the performance of AtacWorks to one existing convolutional network approach from Rai et al. Additional benchmarking with other recently published deep convolutional network approach for single cell datasets, such as SCALE and scVI – which perform denoising and imputation- should be performed to compare performance, (even if these models don't necessarily take as input the same types of data).

Response 2.2: We agree with the reviewer that benchmarking is important, however the methods suggested by the reviewer (SCALE and scVI) use different data and also produce different outputs, and thus comparisons to AtacWorks is not in scope within this work. scVI is designed specifically for single-cell RNA-seq data, and thus a direct comparison would require extensive modification.

SCALE is designed for single-cell ATAC-seq experiments. However, rather than denoising sequencing coverage across the entire genome, SCALE denoises the peaks by cells matrix used for clustering, which is a much lower resolution output. SCALE also does not call peaks, but is limited to a pre-defined set of peaks. This is very different from AtacWorks which produces base pair-resolution signal and also calls peaks. The main purpose of the output produced by SCALE is to improve dimension reduction and clustering of cells. In contrast, we apply AtacWorks to single-cell data downstream of clustering, to take the given clusters and obtain more accurate base pair-resolution signal tracks and peak calls that can be used to characterize and compare them. AtacWorks also allows us to identify novel peaks that may be specific to rare clusters; both tasks that SCALE cannot perform.

We believe these approaches are synergistic rather than competitive with AtacWorks. SCALE could be used to generate accurate, well-separated clusters of cells which can then be characterized and compared at high resolution using AtacWorks.

Comment 2.3: Regarding false positive, for example in Fig. 1f, the reader appreciates by looking at the tracks the denoising of the low-quality datasets by AtacWorks; yet a series of signal bumps appear on the track as well, suggesting that there could be false positive signal on the denoised coverage track. While it does seem to affect the AUPRC/AUPROC metrics, which are based on the peak called by AtacWork, it could be an issue if the coverage track is used for other purposes than peak calling. The authors should further study and comment this issue of the coverage track, and potentially propose solutions to remove such signal.

Response 2.3: We thank the reviewer for this observation. We note the signal bumps that appear on the denoised track produced by AtacWorks in the previous Fig. 1f (now Supplementary Fig. 4b). However, on closer inspection of these ‘signal bumps’ we see that these are not false positives introduced by AtacWorks, but are elevated in the noisy signal as well:

We see that in each case, these regions are elevated in the noisy signal (blue) itself, due to subsampling noise. While AtacWorks removes noise from the rest of the noisy signal, some noise remains in these few regions. Therefore, rather than describing these regions as false positives, we would characterize them as noise that AtacWorks is unable to fully correct.

We note that although AtacWorks is unable to fully remove the signal noise in these regions, it a) does not increase the magnitude of the signal in these regions, either leaving it unchanged or else reducing it, and b) it recognizes that many of these are not true peaks, as seen from the peak calls.

Though we agree with the reviewer that these cases may not be reflected in the AUPRC and AUROC metrics, we can quantitatively analyze the signal tracks themselves using the regression metrics provided in Supplementary Table 6. Compared to the noisy signal, the denoised signal produced by AtacWorks has better Pearson correlation and Spearman correlation relative to the clean signal. This improvement holds true for genomic regions that fall outside of peaks, indicating that AtacWorks improves the shape of the signal track and does not introduce widespread false positive signals.

Comment 2.4: Concerning the validation of the denoising procedure, for example in the case of Fig. 2f, with single-cell cumulative bulks, peaks are discovered by AtacWorks in sub-populations. How could these be experimentally or in silico validated to check that these are not false positives?

Response 2.4: We agree with the reviewer's assessment that some validation of potential regulatory elements is required. The elements we identify are likely not false positives introduced by the model because we filter them by significance, which is calculated from a background distribution of randomly-selected and denoised HSCs. As for whether the elements are truly associated with lineage priming, we have performed new analyses validating that they are more accessible in the corresponding differentiated cell types, copied below.

To validate that these identified regulatory elements are associated with lineage-priming, we confirmed that the lymphoid-primed elements were more accessible in the CD34⁺ cells from lymphoid lineage (Supplementary Fig. 8a), while the erythroid-primed elements were more accessible in CD34⁺ cells from the erythroid lineage (Supplementary Fig. 8b). We also observed that the most differentially-accessible sequence motifs in these two subsets of peaks included transcription factors crucial to differentiation, including E2F³⁰ and MYB families³¹ (Supplementary Table 16).

Comment 2.5: In order to verify that the model is not simply learning features from the training set, the authors show that regions not in the training set (chromosome 10) are successfully denoised from the low-coverage experiment (Fig 1c, d, Supplementary Table 1). Additionally, they show that the results were highly robust to different subsets, when using a series of hematopoietic cells as training datasets (Supplementary Table 3). While these results are a good evidence that the model is not overfitting the training set, it would be very interesting to question more deeply the impact of the training dataset. For instance, we would recommend comparing the results obtained for one test set with a series of very different cell types. Indeed, the vast majority of datasets originate from hematopoietic cells, the authors should add ATAC-seq from a couple of different tissues to challenge the effect of the training set on the performance of the model.

Response 2.5: We agree with the reviewer that it is important to show that an AtacWorks model trained on hematopoietic cells can generalize to non-hematopoietic cells. To address this concern, we have now added an experiment where we take the AtacWorks model trained on single-cell (dscATAC) data from human blood cells, and apply it to denoise and call peaks from dscATAC mouse brain data. These results are presented in Figure 2c and 2d (copied below) and in Supplementary Table 10.

We have also revised portions of the main text, copied below:

We then tested whether these models trained on dscATAC data from human blood could generalize to non-blood cell types. To do this, we obtained single-cell data from a mouse brain using the same dscATAC protocol¹⁷. We then applied the models trained on human blood to data aggregated from mouse pyramidal and oscillatory neurons. For both types of neurons, we observed that AtacWorks improved the signal track and peak calls, both overall and within cell-type specific peaks (Fig. 2c, d, Supplementary Table 10). This result demonstrates that AtacWorks is broadly applicable across both cell types and species.

Comment 2.6: We had several remarks about the usability of AtacWork for single-cell datasets. Could AtacWork be used for peak inference only on individual cells for example? Why do the authors limit their usage of AtacWork on groups of 50 cells precisely? Overall we could expect more insight on the application of AtacWork to scATAC-seq.

Response 2.6: We agree with the reviewer that the original manuscript was unclear on the parameters for single-cell denoising. To address this concern, we now present results for AtacWorks models applied to groups of 1, 5, 10, and 50 cells (Fig. 2b, Supplementary Tables 8 and 9).

Fig. 2b

AUPRC of peak calls on aggregate single-cell ATAC-seq data from human natural killer (NK) cells. Peak calls were produced by MACS2 (blue) and AtacWorks (green) on noisy data aggregated over 1-50 cells. Gray bars show AUPRC of MACS2 on larger numbers of cells, to illustrate how many cells MACS2 requires to reach the same performance as AtacWorks.

We have also revised the main text to describe the improvements in signal tracks and peak calling, as well as the potential limitations of denoising small numbers of cells, copied below:

We randomly sampled 1 cell (~20,000 reads), 5 cells (~0.1 million reads), 10 cells (~0.2 million reads) or 50 cells (~1 million reads) for the low-coverage training datasets. The resulting trained models improved signal track accuracy and peak calling from aggregated natural killer (NK) cells sequenced using the same protocol (Fig. 2b, Supplementary Table 8, Supplementary Table 9). Notably, AtacWorks improved the AUPRC of peak calls from 50 NK cells from 0.2048 to 0.7008, a result that MACS2 requires over 400 cells to obtain (Fig. 2b, Supplementary Table 8). Though we observed improved signal quality and peak calls for any number of cells, the results on 1 and 5 cell samples may be too noisy for downstream biological analysis, possibly due to single-cell heterogeneity not captured by the aggregate data used for training.

Reviewer #3

Comment 3.1: This manuscript presents AtacWorks, a tool for denoising low-coverage and low-quality ATAC data. AtacWorks looks like a well-written piece of software. Furthermore, the paper presents convincing demonstrations of its utility. AtacWorks is an excellent contribution to the community and I believe it will get a lot of use. The paper is well-organized and well-written I have no problem recommending publication of the paper as is.

I have only a few minor suggestions and comments that I offer for your consideration:

Response 3.1: We thank the reviewer for their comment that AtacWorks is “an excellent contribution to the community” and for their helpful suggestions. We address the rest of their comments below, and have colored significant revisions to the manuscript dark blue for clarity.

Comment 3.2: I believe you never actually present results of denoising a single cell individually -- always aggregated into small groups (of 50). Presumably, this is because the single cell is simply too little data for the method. Can you comment on this?

Response 3.2: We agree with the reviewer that the original manuscript was unclear on the parameters for single-cell denoising. To address this concern, we now present results for AtacWorks models applied to groups of 1, 5, 10, and 50 cells (Fig. 2b, Supplementary Tables 8 and 9).

Fig. 2b

AUPRC of peak calls on aggregate single-cell ATAC-seq data from human natural killer (NK) cells. Peak calls were produced by MACS2 (blue) and AtacWorks (green) on noisy data aggregated over 1-50 cells. Gray bars show AUPRC of MACS2 on larger numbers of cells, to illustrate how many cells MACS2 requires to reach the same performance as AtacWorks.

We have also revised the main text to describe the improvements in signal tracks and peak calling, as well as the potential limitations of denoising small numbers of cells, copied below:

We randomly sampled 1 cell (~20,000 reads), 5 cells (~0.1 million reads), 10 cells (~0.2 million reads) or 50 cells (~1 million reads) for the low-coverage training datasets. The resulting trained models improved signal track accuracy and peak calling from aggregated natural killer (NK) cells sequenced using the same protocol (Fig. 2b, Supplementary Table 8, Supplementary Table 9). Notably, AtacWorks improved the AUPRC of peak calls from 50 NK cells from 0.2048 to 0.7008, a result that MACS2 requires over 400 cells to obtain (Fig. 2b, Supplementary Table 8). Though we observed improved signal quality and peak calls for any number of cells, the results on 1 and 5 cell samples may be too noisy for downstream biological analysis, possibly due to single-cell heterogeneity not captured by the aggregate data used for training.

Comment 3.3: It's interesting that AtacWorks can pick up signals not found in the training data. However, this would have been more convincing if you'd have used a non-blood sample for your test case. As is, you trained the model using a variety of different blood cell types, and then your held-out sample was another blood cell type. It would be interesting if this capacity is reduced the further away you get from the training data.

Response 3.3: We agree with the reviewer that it is important to show that an AtacWorks model trained on blood cells can be applied to non-blood cells. To address this concern, we have added an experiment where we take the AtacWorks model trained on single-cell (dscATAC) data from human blood cells, and apply it to denoise and call peaks from the dscATAC mouse brain data. These results are presented in Fig. 2c, d (copied below) and Supplementary Table 10.

We have also revised portions of the main text, copied below:

We then tested whether these models trained on dscATAC data from human blood could generalize to non-blood cell types. To do this, we obtained single-cell data from a mouse brain using the same dscATAC protocol¹⁷. We then applied the models trained on human blood to data aggregated from mouse pyramidal

and oscillatory neurons. For both types of neurons, we observed that AtacWorks improved the signal track and peak calls, both overall and within cell-type specific peaks (Fig. 2c, d, Supplementary Table 10). This result demonstrates that AtacWorks is broadly applicable across both cell types and species.

Comment 3.4: But perhaps more important, I would like to know if you can tease out of the network any information about exactly what features contribute to this ability to predict outside of the training data. I know deep networks are notorious for difficulty of interpretation, but nevertheless, perhaps some feature importance approaches may aid in interpreting the model.

Response 3.4: We agree with the reviewer that it might be useful to interpret the specific features learned by our neural network. We examined the relevant literature and found that feature importance approaches for convolutional signal denoising models such as AtacWorks are not well developed. We were unable to find a simple interpretation method that could be applied or easily adapted to our framework. We believe that useful interpretation of the features of this model would require significant development, which lies outside the scope of our study.

Comment 3.5: For the peak calling component of the model, you state that the classifier operates on each nucleotide individually. How do you prevent the model from learning discontinuous peaks, which are not realistic? Considering each nucleotide independently seems likely to lead to such a scenario, where the peaks are spotty throughout the genome.

Response 3.5: We thank the reviewer for raising this important point. Our model makes predictions on each base individually. However, the prediction is based not only upon the coverage of that individual base, but also upon the coverage of nearby bases on either side. Therefore, the model learns the context in which a true peak occurs, and an isolated base or few bases with high coverage are unlikely to be called a peak.

However, some discontinuous peaks do remain. We filter these out using the ‘macs2 bdgpeakcall’ command from MACS2. This command calls peaks using a user-defined threshold (we use a probability of 0.5), filters out short peaks and combines peaks that are close together. The default parameters are 200 bp for the minimum length of a peak, and 50 bp for the minimum distance between two peaks.

We acknowledge that our peak calling procedure was not clearly described in the original manuscript. We have now clarified this in the Methods section as follows:

Peak calling

In order to call peaks from the base pair-resolution probabilities produced by AtacWorks, the ‘macs2 bdgpeakcall’ command from MACS2¹³ was run with a threshold of 0.5. This is the same procedure used by MACS2 to call peaks from base pair-resolution p-values.

Comment 3.6: For the final analysis with 1000 samples of 50 randomly selected HSCs, it's not clear to me that this is the best background distribution to draw from, as I would expect the denoised tracks to be less useful with 50 random cells than with 50 nearest neighbors to a single cell. It seems a more appropriate analysis would be to select 1000 random single cells, and then choose the 50 nearest neighbors to each, and then use that to build a null distribution.

Response 3.6: We agree with the reviewer's assessment that our original background distribution may not be the most appropriate. To address this concern, we have generated a new background distribution as suggested and have updated the results (which did not change significantly) in the main text, copied below:

To assess the significance of these chromatin accessibility differences, we took 1000 random samples of 50 similar HSCs each and used AtacWorks to denoise a list of select genomic regions (200 kb windows surrounding 2,303 genes with differential expression across CD34⁺ cells, see Methods). For each of the ~300 million bases in the selected regions, we calculated a normalized mean and standard deviation of the coverage from the 1000 denoised tracks, allowing us to estimate z-scores for each regulatory region we observed in our denoised long-term HSC and lineage-primed samples (see Methods). We identified an average of 2,863 regulatory regions with significant differential accessibility per sample (Supplementary Table 15), including those highlighted (Fig. 3f).

Comment 3.7: The github URLs appear to be outdated.

Response 3.7: We thank the reviewer for pointing this out, and have updated the GitHub URLs.

Reviewer #4

Comment 4.1: This paper focuses on improving the signal resolution of ATAC-seq data of chromatin accessibility at single-cell level. The authors introduce a deep learning based model, named AtacWorks, to analyze ATAC-seq data so that biologically-meaningful change could be measured more efficiently. Based on ResNet, AtacWorks integrates three types of optimized function, two for data denoising and one for peak calling. The model was applied to multiple tasks, including denoising single-cell aggregated data, cross-modality prediction, and enhancing the resolution of rare subpopulations of single cells. These results are all validated across different cell types demonstrating the model robustness.

This manuscript is well-written. For further improvement the authors could address the following comments.

Response 4.1: We thank the reviewer for their comment that the manuscript is “well-written” and for their helpful suggestions for further improvement. We address the rest of their comments below, and have colored significant revisions to the manuscript dark blue for clarity.

Major comments

Comment 4.2: While the main contribution lies in the development of a new method, the description of the method looks a bit superficial. Readers can see that AtacWorks has good performance, but there is little insight into why. In particular, why is ResNet more suitable than the U-Net previously used for the tasks here?

Response 4.2: We have added a section discussing this point to Supplementary Note 1, which we quote below:

While we cannot rule out the possibility that a U-Net architecture designed and tuned in a specific way would outperform the ResNet architecture used in AtacWorks, we found that in all of our tests, the ResNet architecture consistently performed best on both denoising and peak calling tasks.

U-Net models have shown excellent performance on a variety of tasks in computer vision. We cannot give a definitive answer as to why we have observed the ResNet architecture to outperform U-Net on this particular application. We note a few possible reasons why the ResNet may be more suitable than the U-Net for the tasks here:

1. The U-Net model contains “max pooling” layers which reduce the size of the data representation by retaining only the maximum value across neighboring units, thus reducing resolution. The ResNet architecture does not include max pooling layers and does not compress the size of the data representation, instead using dilated convolutions to combine information over a large genomic distance.
2. The final layer of the U-Net combines very low-resolution features learned by the first layers of the model with high-resolution features spanning kilobases, which are learned by the final layers. This may not be ideal due to the very different scales of the features being combined. In the ResNet architecture, skip connections only skip every three convolutional layers, thus transferring information from shallower to deeper layers of the model without combining features of drastically different scales.

Please also see Responses 4.3 and 4.7 for additional comparison of the ResNet and U-Net architectures.

Comment 4.3: If the authors of the U-Net method do some parameter tuning, could it outperform AtacWorks in turn? The results comparison with U-Net should be shown more clearly in the results section.

Response 4.3: We thank the reviewer for this suggestion. In our previous results, we had re-implemented the U-Net architecture described in PillowNet with the reported parameter values, and shown that the AtacWorks ResNet model performs better than the PillowNet U-Net model at both denoising and peak calling (Supplementary Note 1, Supplementary Table 7).

To answer the question as to whether a differently tuned U-Net model might outperform the ResNet, we also independently designed and tuned our own U-Net model, which differs from the PillowNet model in several respects (detailed in Supplementary Note 1), and compared it with an AtacWorks ResNet model. Details about this model and its performance are presented in Supplementary Note 1 (copied below) and Supplementary Table 7.

3. Comparison of an AtacWorks ResNet model with an independently designed U-Net

To explore the suitability of U-Net architectures for denoising and peak calling from noisy ATAC-seq data, we also independently designed and tuned a U-Net architecture for these tasks. To be consistent with the framework of AtacWorks, our U-Net architecture differed from the U-Net architecture used in PillowNet in several ways:

- a. We developed a single model for both denoising and peak calling.
- b. Whereas PillowNet uses both the noisy ATAC-seq signal and peak calls from MACS2 as input for classification, our U-Net model uses only the noisy ATAC-seq signal.
- c. Our model was trained with a joint MSE, Pearson correlation and BCE loss function, whereas PillowNet models are trained with MSE loss for regression and BCE loss for classification.

The runtime for training this independent U-Net model using 8 Tesla V100 16GB GPUs was:

- 254 seconds per epoch (regression + classification)

We tuned the hyperparameters of this model based on performance on the validation set consisting of a held-out chromosome. Performance was improved by using wider convolutional filters (25 bp compared to the 11 bp filters used in PillowNet). Nevertheless, the performance of this U-Net model on the task described above was comparable to that of the models based on the PillowNet architecture (Supplementary Table 7), and still did not match the performance of the ResNet model.

Thus, both our independently designed and tuned U-Net architecture, as well as the previously described PillowNet U-Net architecture, perform worse than the ResNet. This suggests that the ResNet is a more suitable choice of architecture for this task.

Comment 4.4: The cross-modality predictions look interesting. However, what is the motivation for doing that, considering that scATAC-seq is probably no more reliable than the data of other modalities? Maybe ATAC-seq could be used for interpreting or complementing other data, or the authors may just want to show the reliability of AtacWorks. Please explain it after line 218.

Response 4.4: We agree with the reviewer that the motivation behind the cross-modality predictions was unclear in the original manuscript. To address this concern, we have revised portions of the main text to include potential use cases of this approach, copied below:

These cross-modality predictions demonstrate the potential for AtacWorks to generate multiple layers of information in single cells from one of the most commonly-used epigenomic assays, at no additional cost. It is generally experimentally challenging to make multiple measurements from the same cells, so this approach may be especially useful in cases where running multiple ChIP-seq experiments is infeasible due to time, reagents, sample availability, or biological variability. Though the models presented here tend to perform better on active histone marks (e.g. H3K27ac) or abundant architectural proteins (e.g. CTCF), these specific predictions may be useful for distinguishing active vs. poised enhancers²¹ or characterizing changes in 3D genome structure across differentiation²². We anticipate future work will extend these capabilities to enable cross-modality inference of additional latent epigenetic states from a single experiment.

Comment 4.5: From the perspective of learned patterns or model performance, please explain the model's generalization on unseen cell types.

Response 4.5: We have revised the Introduction and Discussion sections to cover this topic:

Introduction:

The network makes predictions for each base in the genome based on coverage values from a surrounding region spanning several kilobases (6 kb for the models presented here), but does not consider the DNA sequence itself, allowing it to generalize across cell types.

Discussion:

AtacWorks is not provided with the DNA sequence as an input, which means it is agnostic to cell- or condition- specific correlations between chromatin accessibility and sequence motifs. Instead, the model learns features based on the height and shape of the coverage track, which generalize across datasets.

Minor Comments

Comment 4.6: The training datasets had better include at least one collected from real experiments rather than only subsampling from high-converge dataset.

Response 4.6: The purpose of the training datasets is to enable the model to learn an accurate mapping from noisy ATAC-seq data to corresponding higher-coverage or higher-quality data. We preferred to avoid using two real datasets from separate experiments to learn this mapping because the model would then suffer from the effects of technical parameters such as uneven sample quality, unequal mixtures of cell types, cell cycle variability, etc. Hence we opted to use random subsampling to create noisy datasets, which allowed us to limit experimental artifacts because the pairs of clean and noisy datasets seen by the model correspond to one another.

In case the reviewer's comment is referring to the low-coverage testing datasets used to evaluate the performance of the model, these low-coverage datasets are subsampled in order to accurately measure the performance of AtacWorks. Since the low-coverage datasets are generated directly from high-coverage datasets, we can compare the denoised track and peak calls by AtacWorks to the high-coverage track and peak calls to calculate performance metrics. As before, if the low-coverage testing dataset was not generated

directly from high-coverage data but was instead taken from a separate experiment, technical differences between the two experiments could lead to inaccurate performance metrics.

Comment 4.7: The runtime comparison with U-Net is unfair, since U-Net was implemented on CPU while AtacWorks is on GPU. For a fair comparison, the U-Net-based model should also be trained on GPU with the same environment as AtacWorks.

Response 4.7: Our intention in the original comparison was simply to compare the runtime and usability of our software with the software released in the previous study. We agree with the reviewer that it would also be useful to provide the runtime of the two model architectures in the same environment. Accordingly, we have performed the comparison suggested by the reviewer and presented the results in Supplementary Note 1 (copied below) and Supplementary Table 7.

Since we were unable to apply the PillowNet code to chromosome-scale data, we instead re-implemented the U-Net architecture used in PillowNet in the AtacWorks framework. This allowed us to solely compare the default model architectures used by AtacWorks and PillowNet, in the same environment.

We were able to train U-Net models for denoising and peak calling on the aforementioned CD4+ T cell dataset with the U-Net architecture, using the loss functions and learning rate described¹. We also trained a standard AtacWorks ResNet model to perform both denoising and peak calling using default AtacWorks parameters (Supplementary Table 17).

The runtime for training using 8 Tesla V100 16GB GPUs was:

- AtacWorks ResNet: 185 seconds per epoch (regression + classification)
- U-Net (PillowNet reimplementation): 168 seconds per epoch (regression model) + 168 seconds per epoch (classification model)
-

We then applied the trained models to an ATAC-seq dataset from erythroblasts sampled to the same read depth.

We found that while the U-Net architecture performs well at both denoising and peak calling from this low-coverage dataset, the ResNet model performs better on all metrics (Supplementary Table 7).

We show that the ResNet architecture of AtacWorks outperforms a U-Net architecture in both speed and performance, even when both models are trained in the same environment.

Comment 4.8: For “potential for AtacWorks to be broadly applied for cross-modality inference of latent epigenetic states”, the authors could have more analysis on either model training or results to demonstrate this potential.

Response 4.8: We agree with the reviewer’s assessment that our cross-modality inference demos do not necessarily demonstrate broad application. Though we remain excited about the future potential of AtacWorks to broadly perform cross-modality predictions, we have revised portions of the main text to more accurately reflect the limitations of the current approach, see Response 4.4.

Comment 4.9: The full spelling of “HSCs” should be given when the word firstly appears in the article.

Response 4.9: We thank the reviewer for pointing this out and have included the full spelling when it first appears, copied below:

We then downsampled these tracks to lower sequencing depths and trained a model for each depth, which we tested on data from similarly-processed hematopoietic stem cells (HSCs).

Comment 4.10: As stated in the paper, the so-called “potential regulatory elements” have not been validated, therefore could only be called “novel signals” since they may also be artifacts due to some bias introduced by the predictive model. Some validation or analysis should be given here.

Response 4.10: We agree with the reviewer’s assessment that some validation of potential regulatory elements is required. The elements we identify are likely not false positives introduced by the model because we filter them by significance, which is calculated from a background distribution of randomly-selected and denoised HSCs. As for whether the elements are truly associated with lineage priming, we have performed new analyses validating that they are more accessible in corresponding differentiated cell types, copied below.

To validate that these identified regulatory elements are associated with lineage-priming, we confirmed that the lymphoid-primed elements were more accessible in the CD34⁺ cells from lymphoid lineage (Supplementary Fig. 8a), while the erythroid-primed elements were more accessible in CD34⁺ cells from the erythroid lineage (Supplementary Fig. 8b). We also observed that the most differentially-accessible sequence motifs in these two subsets of peaks included transcription factors crucial to differentiation, including E2F³⁰ and MYB families³¹ (Supplementary Table 16).

REVIEWERS' COMMENTS

Reviewer #1 (Remarks to the Author):

In the revised manuscript the authors clarify a number of items that full address my major concerns. Specifically, that any ATAC data can be used to train a model to be applied to any other dataset. The fig 2a highlights this and while I am still a little skeptical on the ability o pick up peaks not present in the training datasets – the authors report on exactly that and in multiple contexts. They also now include far more datasets, which is valuable for demonstrating versatility.

I see AtacWorks as being applicable to a wide variety of projects – particularly with the availability of single-cell ATAC-seq. I also think it is worth noting that the current analysis workflows for ATAC-seq are to use the default macs2 peak caller, which was never even designed for ATAC-seq data. While it works well, addressing peak calling for ATAC-seq is an area that I believe is important, and something that this work contributes to.

Reviewer #2 (Remarks to the Author):

The authors have made a great job at answering my concerns, and clarifying the manuscript, which now ready for publication in my view.

Reviewer #4 (Remarks to the Author):

The authors have made satisfactory responses to my comments and extensive revisions. Hence I recommend this manuscript to be accepted.

Reviewer #1

Comment 1.1: In the revised manuscript the authors clarify a number of items that full address my major concerns. Specifically, that any ATAC data can be used to train a model to be applied to any other dataset. The fig 2a highlights this and while I am still a little skeptical on the ability to pick up peaks not present in the training datasets – the authors report on exactly that and in multiple contexts. They also now include far more datasets, which is valuable for demonstrating versatility.

Response 1.1: We thank the reviewer for highlighting the lack of clarity around data requirements in the first version of the manuscript. We are glad we were able to address this.

Comment 1.2: I see AtacWorks as being applicable to a wide variety of projects – particularly with the availability of single-cell ATAC-seq. I also think it is worth noting that the current analysis workflows for ATAC-seq are to use the default macs2 peak caller, which was never even designed for ATAC-seq data. While it works well, addressing peak calling for ATAC-seq is an area that I believe is important, and something that this work contributes to.

Response 1.2: We also agree with the suggestion regarding MACS2, and have now mentioned this fact in the main text as follows:

Peaks for each clean dataset were identified using MACS2¹⁴ (see Methods) which is the standard peak caller for ATAC-seq data, despite not being developed specifically for that purpose.

Reviewer #2

Comment 2.1: The authors have made a great job at answering my concerns, and clarifying the manuscript, which is now ready for publication in my view.

Response 2.2: We thank the reviewer for their constructive comments which have helped us improve the manuscript.

Reviewer #4

Comment 4.1: The authors have made satisfactory responses to my comments and extensive revisions. Hence I recommend this manuscript to be accepted.

Response 4.1: We thank the reviewer for their constructive comments which have helped us improve the manuscript.